# Demographic Parity Constrained Minimax Optimal Regression under Linear Model

**Kazuto Fukuchi**[1,3*]   **Jun Sakuma**[2,3]
[1] University of Tsukuba    [2] Tokyo Institute of Technology    [3] RIKEN
fukuchi@cs.tsukuba.ac.jp
sakuma@c.titech.ac.jp

## Abstract

We explore the minimax optimal error associated with a demographic parity-constrained regression problem within the context of a linear model. Our proposed model encompasses a broader range of discriminatory bias sources compared to the model presented by Chzhen and Schreuder [6]. Our analysis reveals that the minimax optimal error for the demographic parity-constrained regression problem under our model is characterized by $\Theta(dM/n)$, where $n$ denotes the sample size, $d$ represents the dimensionality, and $M$ signifies the number of demographic groups arising from sensitive attributes. Moreover, we demonstrate that the minimax error increases in conjunction with a larger bias present in the model.

## 1   Introduction

Machine learning techniques have been incorporated into numerous automated decision-making systems, spanning critical domains such as employment, credit assessment, insurance, and security. Nevertheless, these systems can exhibit discriminatory behavior towards specific demographic groups, including gender, race, and ethnicity, potentially causing significant societal ramifications. This issue, known as the fairness problem, has attracted substantial attention within the machine learning research community. The growing focus on the fairness problem primarily arises from reported instances of unfair behavior in real-world systems, encompassing recidivism risk prediction [3], hiring practices [10], facial recognition [9, 17], and credit scoring [24].

Motivated by these concerns, a considerable body of research has explored regression problems subject to fairness constraints [13, 25, 16, 2, 7, 8, 14, 18]. Numerous regression algorithms incorporating various fairness constraints have been developed to accommodate diverse contexts, with demographic parity [21] and equalized odds [11] being the predominant fairness constraints adopted by these methods.

In this study, we focus on the regression problem under the fairness constraint of demographic parity [21]. Existing literature primarily concentrates on the development of fair regression algorithms, and their performance evaluation predominantly relies on empirical analyses. Such evaluations, however, only offer performance guarantees for specific scenarios explored in the experiments, which may result in poor performance in unexamined situations. To ensure the algorithm's robust performance across a wider range of contexts and obtain a comprehensive understanding of the fair regression problem, a theoretical analysis of statistical efficiency is indispensable.

Several studies have introduced fair regression algorithms accompanied by theoretical analyses of their statistical efficiency in terms of accuracy and fairness. Agarwal et al. [2] designed a demographic parity-based fair regression algorithm using reduction methods [1] and established upper bounds on its empirical approximation errors for accuracy and fairness using Rademacher complexity. Chzhen

---

[*]Corresponding author.

37th Conference on Neural Information Processing Systems (NeurIPS 2023).

Table 1: Comparison between Chzhen and Schreuder [6]'s and our models. Each checkmark signifies the presence of an influence exerted by the sensitive attribute on the respective variable.

| | partial coefficients | intercept | non-sensitive features |
|---|---|---|---|
| Chzhen and Schreuder [6] | | ✓ | |
| ours | ✓ | ✓ | ✓ |

et al. [8] proposed a discretization-based fair regression algorithm, deriving upper bounds on the mean squared excess risk for accuracy and a Kolmogorov distance-based score for demographic parity as fairness guarantees. Chzhen et al. [7] derived the Bayes optimal regressor under a demographic parity constraint, providing upper bounds on the mean absolute deviation from the Bayes optimal regressor for accuracy and a Kolmogorov distance-based score for demographic parity as fairness guarantees. Despite ensuring low error and fair treatment even in non-linear models, it remains unclear if these guarantees represent optimal performance among possible algorithms.

**Minimax optimal fair regression.** Numerous researchers have investigated minimax optimal regression algorithms, the best possible algorithm, without addressing fairness considerations [22, 23, 19, 15]. In contrast to standard regression problems, minimax optimality in fair regression problems remains relatively unexplored, with a notable exception being the recent work by Chzhen and Schreuder [6]. They examine minimax optimality in fair regression problems, incorporating demographic parity constraints within the following linear model:

$$Y = \langle \beta^*, X \rangle + b_S + \xi \text{ where } X \sim N(0, \Sigma). \tag{1}$$

In this model, $Y$, $X$, and $S$ represent the outcome, non-sensitive features, and a sensitive attribute, respectively. $\langle \cdot, \cdot \rangle$ denotes the inner product, $\xi$ represents zero-mean noise, and $\Sigma$ is an arbitrary covariance matrix. For example, in salary calculations, $X$ and $S$ correspond to working hours and gender, respectively, with $b_S$ and $\beta^*$ signifying the base salary and hourly wage. In Eq (1), the salary $Y$ is determined by the base salary $b_S$ and the product of working hours $X$ and an hourly wage $\beta^*$.

The model in Eq (1) exhibits limitations pertaining to its applicability across various scenarios. We elucidate these limitations by discussing the notion of *direct discrimination* and *indirect discrimination*, summarized succinctly in the second row of Table 1. Direct discrimination occurs when the sensitive attribute influences the outcome, regardless of non-sensitive features. The model in Eq (1) can treat direct discrimination resulting from the dependency of the intercept $b_S$ on $S$; for example, it can capture discrimination due to basing base salary on gender (third column in Table 1). However, it is imperative to underscore that the model in Eq (1) fails to handle direct discrimination arising from the partial (regression) coefficients $\beta^*$, as these are independent of $S$; for instance, it cannot accommodate discrimination due to gender-dependent hourly wages (second column in Table 1).

Indirect discrimination (or redlining effect [4]) constitutes another source of unfair bias, arising when the sensitive attribute influences the outcome through its correlation with non-sensitive features. The presence of the dependency between non-sensitive features and the sensitive attribute signifies indirect discrimination. In the model in Eq (1), non-sensitive features $X$ is independet from the sensitive attribute $S$, thereby implying an absence of indirect discrimination (forth column in Table 1).

Chzhen and Schreuder [6] effectively revealed the minimax optimal error for fair regression problems involving direct discrimination due to varying intercepts associated with sensitive attributes. However, their research does not address direct discrimination from partial coefficients and indirect discrimination through non-sensitive features.

**Our model and contributions.** In this study, we investigate the minimax optimality of the fair regression problem in the context of the following model:

$$Y = \langle \beta_S^*, X \rangle + \xi \text{ where } X \sim N(\mu_S, \sigma_X I), \tag{2}$$

where $\sigma_X > 0$, and $I$ denotes the identity matrix. The subscript in $\beta_S^*$ and $\mu_S$ signifies that our model varies regression coefficients and the mean of non-sensitive features based on the sensitive attribute.

Compared to the model proposed by Chzhen and Schreuder [6], our model accommodates a broader range of direct and indirect discrimination. These discriminations can be characterized as follows:

- (Direct discrimination) Our model accommodates direct discrimination through discrepancies in $\beta_S^*$ concerning $S$, as the regression coefficients $\beta_S^*$ hinge on the sensitive attribute $S$ (second and third columns on the third row in Table 1). This includes, for instance, discrimination arising from varying base salaries and hourly wages. Divergent partial coefficients yield varied outcome variance amongst $S$, while disparate intercepts relative to $S$ merely alter the outcome's mean. Hence, our model introduces an additional challenge of attenuating direct discrimination through disparate variance, alongside mitigating direct discrimination through disparate mean. This presents a stark contrast to Chzhen and Schreuder [6]'s model, which solely focuses on mitigating discrimination via the mean without considering the variance.
- (Indirect discrimination) The sensitive attribute $S$ affects the mean of non-sensitive features $X$, as denoted by the subscript of $\mu_S$. Our model thereby introduces indirect discrimination through variations in $\mu_S$ with respect to $S$ (e.g., disparate working hours by gender). To alleviate this form of indirect discrimination, $\mu_S$ needs to be estimated to adjust the learned regressor, thereby ensuring its output remains invariant to differing $\mu_S$. Therefore, our model presents an additional complexity in estimating $\mu_S$ for mitigating indirect discrimination.

Overall, our model demonstrates an expanded dependency of partial coefficients (direct discrimination) and non-sensitive features (indirect discrimination) on the sensitive attribute (second and fourth columns of Table 1).

The principal contribution of this paper lies in the establishment of matching upper and lower bounds on the minimax optimal error (i.e., the error corresponding to the minimax optimal regression algorithm) and the proposition of a regression algorithm that achieves this optimal error under Eq (2). The optimal error elucidates several insights:

- (Direct discrimination) The optimal error comprises a term reflecting the outcome's variance heterogeneity but excludes that of the outcome's mean. This insight implies that mitigating direct discrimination due to the outcome's variance sacrifices statistical efficiency, whereas addressing direct discrimination due to the outcome's mean does not entail this cost. This term effectively quantifies the cost of mitigating direct discrimination in variance and is absent from the optimal error of the Chzhen and Schreuder [6]'s model. Its identification, thus, signifies a crucial contribution of our research.
- (Indirect discrimination) Our lower bound is independent of the term associated with indirect discrimination. Although this evidence is not definitive, it hints at the potential for mitigating indirect discrimination without additional costs under certain conditions. This observation sets the stage for future research focused on developing cost-effective strategies to tackle indirect discrimination.

Our technical contributions to establish these bounds are detailed in Section 4.

**Notations.** Given a positive integer $m$, define $[m] = \{1, ..., m\}$. For a finite set $A$, denote its cardinality by $|A|$. Given an event $\mathcal{E}$, its complement is represented as $\mathcal{E}^c$, and its probability is denoted by $\mathbb{P}\{\mathcal{E}\}$. For a random variable $X$, its expectation is $\mathbf{E}[X]$, and its associated sigma-algebra is $\sigma(X)$. For two real values $a$ and $b$, the notations $a \vee b = \max\{a, b\}$ and $a \wedge b = \min\{a, b\}$ are used. For a square matrix $A \in \mathbb{R}^{d \times d}$, its maximum and minimum eigenvalues are denoted by $\lambda_{\max}(A)$ and $\lambda_{\min}(A)$, respectively, and its transpose is represented by $A^\top$. The set of unit vectors is given by $\mathbb{S}_{d-1}$. For a sequence $a_t$ indexed by $t \in \mathcal{T}$, the notation $a.$ denotes the sequence $(a_t)_{t \in \mathcal{T}}$.

## 2 Problem Setup

### 2.1 Model and Learning Algorithm

**Model.** The proposed model, described in the introduction, is formulated according to Eq (2). We consider $X \in \mathbb{R}^d$ and $S \in [M]$ where $M \geq 2$. The noise variable, $\xi$, is assumed to follow a Gaussian distribution with zero mean and variance $\sigma_\xi^2 > 0$. We define $p_s = \mathbb{P}\{S = s\}$ for all $s \in [M]$, and the optimal regression function is denoted as $f^*(x, s) = \langle \beta_s^*, x \rangle$.

**Learning algorithm.** Given $n$ i.i.d. copies of the tuple $(X, S, Y)$, denoted as $D_n = \{(X_1, S_1, Y_1), ..., (X_n, S_n, Y_n)\}$, the goal is to construct a regression function $f$ that maps $(X, S)$ to $Y$, represented as $\hat{f}_n$. The learner seeks to optimize the accuracy of $\hat{f}_n$ while satisfying a fairness constraint. The definitions of fairness and accuracy are provided in subsequent subsections.

## 2.2 Fairness

**Demographic parity.** We utilize demographic parity [21] as our fairness criterion. A regressor $f$ adheres to demographic parity if its output distribution is invariant when conditioned on $S = s$.

**Definition 1.** *A regressor $f$ satisfies (strong) demographic parity if, for all $s, s' \in [M]$, and for all $E \in \sigma(f(X, S))$, $\mathbb{P}\{f(X, S) \in E | S = s\} = \mathbb{P}\{f(X, S) \in E | S = s'\}$.*

Denote the set of all regressors fulfilling demographic parity for a given distribution of $X$, parameterized by $\mu_{\cdot}$, as $\mathcal{F}_{\mathrm{DP}}(\mu\cdot)$.

**Fairness consistency.** Instead of enforcing strict demographic parity (Definition 1), which results in the regressor to be a constant function due to the unknown $(X, S)$ distribution, we introduce *fairness consistency* (Definition 2). This concept demands the learned regressor to converge to a fair regressor as the sample size $n$ approaches infinity.

To define "convergence", we introduce the *unfairness score $U(f) \geq 0$*, where a lower $U(f)$ indicates a higher fairness level. $U(f) = 0$ if and only if $f$ achieves demographic parity (Definition 1). We claim the learned regressor $\hat{f}_n$ converges to an exactly fair regressor when $U(\hat{f}_n) \to 0$ as $n \to \infty$.

**Definition 2.** *A learning algorithm is $(\alpha, \delta)$-consistently fair for an unfairness score $U$ if there exist constants $n_0 \geq 0$ and $C > 0$, independent of $n$, such that $\mathbb{P}\{U(\hat{f}_n) > Cn^{-\alpha}\} \leq \delta$ for all $n \geq n_0$, with randomness arising from the training sample via $\hat{f}_n$.*

Note that an $(\alpha, \delta)$-consistently fair regressor $\hat{f}_n$ exhibits $(\alpha', \delta)$-consistent fairness for any $\alpha' \in (0, \alpha]$.

We adopt a specific unfairness score using the Wasserstein distance. Given two probability measures $\nu$ and $\nu'$ over $\mathbb{R}$, $\Pi(\nu, \nu')$ denotes the set of all coupling measures $\pi$ satisfying $\pi(A \times \mathbb{R}) = \nu(A)$ and $\pi(\mathbb{R} \times A') = \nu'(A')$ for every measurable sets $A, A' \subset \mathbb{R}$. The 2-Wasserstein distance $W_2$ between $\nu$ and $\nu'$ is expressed as $W_2^2(\nu, \nu') = \inf_{\pi \in \Pi(\nu, \nu')} \int (z - z')^2 \pi(dz, dz')$. Our unfairness score is then formulated as:

$$U(f) = \max_{s, s' \in [M]} W_2(\nu_{f|s}, \nu_{f|s'}),$$

where $\nu_{f|s}$ represents the distribution of $f(X, S)$ conditioned on $S = s$. Prior works, including [2, 7, 8, 6], have adopted different unfairness scores (see the appendix for details[2]).

## 2.3 Accuracy

Under the fairness consistency constraint, the learner's objective is to obtain a fair approximation of $f^*$, denoted as $f_{\mathrm{DP}}^*$, which is the closest regressor to $f^*$ within $\mathcal{F}_{\mathrm{DP}}(\mu_{\cdot})$ using the $L^2$ distance:

$$f_{\mathrm{DP}}^* = \underset{f \in \mathcal{F}_{\mathrm{DP}}(\mu_{\cdot})}{\arg\min} \mathbf{E}\Big[(f(X, S) - f^*(X, S))^2\Big].$$

To evaluate the inaccuracy of a regressor $f$, we compute the mean squared deviation from $f_{\mathrm{DP}}^*$:

$$\mathcal{E}(f; \beta^*, \mu_{\cdot}) = \mathbf{E}\Big[(f(X, S) - f_{\mathrm{DP}}^*(X, S))^2\Big]. \tag{3}$$

Chzhen et al. [7, 8] employ similar definitions, differing only in the choice of deviation metric.

This paper aims to identify the minimax optimal regression algorithm, which minimizes Eq (3) while maintaining fairness consistency. Given parameters $\alpha > 0$ and $\delta \in (0, 1)$, the optimal error is formulated as:

$$\mathcal{E}_n(\alpha, \delta) = \inf_{\hat{f}_n : (\alpha, \delta)\text{-consistently fair}} \sup_{\beta^* \in \mathcal{B}, \mu_{\cdot} \in \mathcal{M}} \mathbf{E}[\mathcal{E}(\hat{f}_n; \beta^*, \mu_{\cdot})],$$

where the infimum is taken over all $(\alpha, \delta)$-consistently fair algorithms, and $\mathcal{B}$ and $\mathcal{M}$ represent the sets of possible $\beta^*$ and $\mu_{\cdot}$, respectively.

---

[2]A version of this paper including appendices is available in the supplementary material.

## 3 Main Results

Our main result is to establish the minimax optimal error bound, delineating the dependency on the diversity of conditional outcome variances concerning the sensitive attribute. This diversity of the variances is quantified via a parameter $B > 0$, which is defined such that it satisfies:

$$\max_s \|\beta_s^*\| \leq B \text{ and } \frac{(\sum_s p_s \|\beta_s^*\|)^2}{M} \sum_s \frac{1}{\|\beta_s^*\|^2} \leq B^2. \tag{4}$$

The left-hand side of the second inequality in Eq (4) forms as a product of two factors: the weighted average norms, $(\sum_s p_s \|\beta_s^*\|)^2$, and the averaged inverse norms, $\frac{1}{M} \sum_s \frac{1}{\|\beta_s^*\|^2}$. As the norms increase, the first factor (weighted average norms) has the propensity to grow, while the second factor (averaged inverse norms) tends to rise when the norms decrease. Maximizing the product of these two elements involves a delicate balancing act: the norms of some groups need to be large, while the norms of other groups need to be smaller. As such, the left-hand side of the second inequality in Eq (4) can increase when the norms $\|\beta_s^*\|$ display diversity.

We adopt mild assumptions on $\beta^* \cdot$ and $\mu \cdot$. Let $\mathcal{B}$ denote the set of $\beta^*$ satisfying Eq (4). Assume there exists a finite universal constant $U > 0$ such that $\|\mu_s\| \leq U$ for all $s \in [M]$, leading to $\mathcal{M} = \{\mu. \in \mathbb{R}^{d \times M} : \forall s \in [M], \|\mu_s\| \leq U\}$. Our analysis relies on these assumptions.

Our main results are as follows:

**Theorem 1.** *Given $\alpha \in (0, 1/2]$ and $\delta \in (0, 1)$, suppose $M(d-1) > 16$ and $n \geq 12(3d \vee 4\ln(M/\delta))/\min_{s \in [M]} p_s$. Then, there exist universal constants $C > 0$ and $c > 0$ such that*

$$c\frac{\sigma_\xi^2 B^2 dM}{n} - o\left(\frac{1}{n}\right) \leq \mathcal{E}_n(\alpha, \delta) \leq C\frac{\sigma_\xi^2 B^2 dM \vee \sigma_X^2 B^2 M \vee B^2 U^2}{n} + o\left(\frac{1}{n}\right).$$

Theorem 1 illustrates that the optimal error is $\sigma^2 \xi B^2 dM/n$ up to a constant factor which may potentially depend on $\sigma_X$ and $U$. The implications of Theorem 1 can be summarized as follows:

1. The optimal error for the standard linear regression problem can be denoted as $d/n$ [15]. The dependency on $n$ and $d$ is consistent with the standard case, provided $\alpha \in (0, 1/2]$.
2. The term $dM$ denotes the number of unknown parameters in Eq (2), comprising $\beta_1^*, .., \beta_M^* \in \mathbb{R}^d$ and $\mu_1, ..., \mu_M \in \mathbb{R}^d$. This dependency on the number of unknown parameters is a common characteristics observed in statistical estimation problems.
3. (**Direct discrimination**) The minimax error delineated in Theorem 1 demonstrates a dependency on parameter $B$. As the variation of $\|\beta_s^*\|$ with respect to $s$ increases, so does the magnitude of $B$. Hence, $B$ serves as a measure of the difficulty in mitigating direct discrimination due to the outcome's variance. This unique quantification of difficulty is absent in standard regression problems and specific to fair regression problems.
4. (**Indirect discrimination**) The lower bound precludes parameters associated with indirect discrimination. It is conceivable that biases arising from indirect discrimination can be reduced without extra costs, provided the dependence of $X$ on $S$ exists only in its mean. Investigating and clarifying this aspect offers a promising direction for future research.
5. The minimax error is invariant to $\alpha$ and $\delta$, implying that the learning process does not introduce unfair bias for $\alpha \in (0, 1/2]$. However, the case for $\alpha \geq 1/2$ remains unexplored and poses a significant research challenge.
6. The gap between the upper and lower bounds regarding $\sigma_X$ and $U$ remains, making narrowing this gap an essential future research direction.

**Remark 1.** *Direct comparison of the minimax error between our model and that of Eq (1) is not feasible due to the differing $f_{DP}^*$ across the models. However, the emergence of the fairness-specific term $B$ can be unequivocally identified as a novel contribution in our study. Notably, the minimax error validated by Chzhen and Schreuder [6] is congruent with the minimax optimal error of standard linear regression within their model, a contrast to our findings.*

To prove Theorem 1, we initiate by constructing the estimator detailed in Section 5. We then prove in Section 6 that the estimator satisfies 1) $(\alpha, \delta)$-fairness consistency for $\alpha \in (0, 1/2]$, and 2) the error aligns with the upper bound specified in Theorem 1. Subsequently, we present a sketch of the proof for the lower bound in Theorem 1 in Section 7. All omitted proofs can be found in the appendices.

# 4 Technical Difficulties in Minimax Optimality Analyses

In this section, we expound on the challenges arising from the analysis of minimax optimality for our problem. First, we introduce the closed-form expression for the Bayes optimal fair regressor $f_{\text{DP}}^*$. We then outline the technical difficulties encountered during the analysis.

**Bayes optimal fair regressor under Eq (2).** Chzhen et al. [7] present a characterization of regression error and the corresponding regressor minimizing the mean squared error under the demographic parity constraint. Building upon the results from Chzhen et al. [7], we derive the closed-form expression for $f_{\text{DP}}^*$ in the following lemma.

**Lemma 1.** *Given the model in Eq (2), the Bayes optimal regressor adhering to the demographic parity constraint can be formulated as*

$$f_{\text{DP}}^*(x, s) = \overline{\|\beta_\cdot^*\|}\left\langle \frac{\beta_s^*}{\|\beta_s^*\|}, x - \mu_s \right\rangle + \sum p_{s'}\langle \beta_{s'}^*, \mu_{s'}\rangle, \tag{5}$$

*where* $\overline{\|\beta_\cdot^*\|} = \sum_{s \in [M]} p_s \|\beta_s^*\|$.

**Technical difficulty in deriving the upper bound in Theorem 1.** To obtain the upper bound in Theorem 1, we first construct an estimator for the regression function in Eq (5) and analyze its regression error. This entails developing estimators for individual components in Eq (5) (e.g., $\overline{\|\beta_\cdot^*\|}$, $\beta_s^*/\|\beta_s^*\|$, $\mu_s$, etc.) and substituting them into Eq (5). The upper bound on $\mathcal{E}_n(\alpha, \delta)$ is derived by combining estimation error bounds for each component's estimator. However, to our best knowledge, no existing estimators provide bounds for the norm ($\|\beta_\cdot^*\|$) and direction ($\beta_s^*/\|\beta_s^*\|$) of regression coefficients. A direct approach involves computing the norm and direction of the OLS estimator, but standard analyses for OLS do not yield bounds on the estimation errors.

The main challenge in deriving the upper bound of Theorem 1 lies in analyzing the following problem: given $X$ following a non-isotropic Gaussian distribution with mean $\mu$, find upper bounds on $\mathbf{E}[(X/\|X\| - \mu/\|\mu\|)^2]$ and $\mathbf{E}[(\|X\| - \|\mu\|)^2]$. Solving this problem provides estimation errors for the norm and direction estimators, as the OLS estimator is an unbiased estimator with noise following the non-isotropic Gaussian distribution. Our key technical contribution is the derivation of these bounds (Theorems 4 and 5).

**Technical difficulty in deriving the lower bound in Theorem 1.** The minimax optimal error characterizes the intrinsic complexity of the regression problem, as no algorithm can surpass this error. In our analysis of the lower bound presented in Theorem 1, we demonstrate that the fair regression problem's complexity, under the model Eq (2), is characterized by the complexity in estimating the direction $\beta_s^*/\|\beta_s^*\|$. The primary challenge lies in establishing this characterization.

To overcome this challenge, we investigate the geometric structure of the error term $\mathcal{E}_n(f; \beta_\cdot^*, \mu_\cdot)$ concerning the parameters $\beta_\cdot^*$ and $\mu_\cdot$. We then reveal that the geometric structure of $\mathcal{E}_n(f; \beta_\cdot^*, \mu_\cdot)$ is characterized by the geometric structure of the direction $\beta_s^*/\|\beta_s^*\|$ (Theorem 7).

# 5 Estimator

In this section, we present a detailed construction of the estimators that attain the minimax error as delineated in Theorem 1. Existing theoretical results, such as those found in Agarwal et al. [2], Chzhen et al. [7, 8], are incapable of addressing unbounded non-sensitive features $X$ or unbounded outcomes $Y$, rendering them inapplicable to our problem. Consequently, we have developed a novel estimator accompanied by rigorous analytical techniques.

**Estimator construction.** In constructing the optimal regressor for model Eq (2), we leverage the results from Lemma 1 and employ a plugin estimator. The method involves estimating the components of terms in Eq (5) and substituting the obtained estimates into the same equation. Concretely, we derive estimators $\widehat{\|\beta_\cdot\|}$, $\tilde{\beta}_s$, $\hat{\mu}_s$, $\hat{p}_s$, $\hat{\beta}_s'$, and $\hat{\mu}_s'$, with the following correspondence:

$$\underbrace{\overline{\|\beta_\cdot^*\|}}_{\widehat{\|\beta_\cdot\|}}\left\langle \underbrace{\frac{\beta_s^*}{\|\beta_s^*\|}}_{\tilde{\beta}_s}, x - \underbrace{\mu_s}_{\hat{\mu}_s} \right\rangle + \sum_{s' \in [M]} \underbrace{p_{s'}}_{\hat{p}_{s'}}\left\langle \underbrace{\beta_{s'}^*}_{\hat{\beta}_{s'}'}, \underbrace{\mu_{s'}}_{\hat{\mu}_{s'}'} \right\rangle.$$

Table 2: Estimator construction. In this table, $\hat{\beta}_{b,s}$ and $\hat{\beta}'_{b,s}$ denote OLS estimands obtained from subsets $D_{b,s}$ and $D'_{b,s}$, respectively. "Sample" refers to the subset utilized for estimand calculation, while "Definition" provides the corresponding estimator's definition. "Sample" in $\widehat{\|\beta.\|}$ is left empty, as it is derived from $\hat{p}_s$ and $\widehat{\|\beta_s\|}$.

| Estimator | Sample | Definition |
|---|---|---|
| $\hat{p}_s$ | $n.$ | $\hat{p}_s = n_s/n$ |
| $\widehat{\|\beta_s\|}$ | $D_{1,s}$ | $\widehat{\|\beta_s\|} = \|\hat{\beta}_{1,s}\|$ if $n_s > 18d$, and $\widehat{\|\beta_s\|} = 0$ otherwise |
| $\widehat{\|\beta.\|}$ | - | $\widehat{\|\beta.\|} = \sum_{s \in [M]} \hat{p}_s \widehat{\|\beta_s\|}$ |
| $\tilde{\beta}_s$ | $D_{2,s}$ | $\tilde{\beta}_s = \hat{\beta}_{2,s}/\|\hat{\beta}_{2,s}\|$ if $n_s > 18d$, and $\tilde{\beta}_s = 0$ otherwise |
| $\hat{\mu}_s$ | $D_{3,s}$ | $\hat{\mu}_s = \frac{1}{n_{3,s}} \sum_{i=1}^{n_{3,s}} X_{3,s,i}$ |
| $\hat{\beta}'_s$ | $D'_{1,s}$ | $\hat{\beta}'_s = \hat{\beta}'_{1,s}$ if $n_s > 12d$, and $\hat{\beta}'_s = 0$ otherwise |
| $\hat{\mu}'_s$ | $D'_{2,s}$ | $\hat{\mu}'_s = \frac{1}{n'_{2,s}} \sum_{i=1}^{n'_{2,s}} X'_{2,s,i}$ |

For technical reasons, we partition the sample to calculate each estimand. Each estimator is assigned a corresponding subset, as shown in Table 2. Under specific conditions, $n_s > 18d$ or $n_s > 12d$, estimators may exhibit altered behavior, primarily as technical considerations for subsequent analyses. We detail the partitioning process as follows. First, we create a histogram of the sensitive attribute $S_i$, denoted as $n. = (n_1, ..., n_M)$, with $n_s = |\{i \in [n] : S_i = s\}|$. Simultaneously, we form group-wise samples $D_s = \{(X_i, Y_i) : i \in [n], S_i = s\}$. For each $s \in [M]$, we partition $D_s$ into $D_{1,s}, D_{2,s}$, and $D_{3,s}$, ensuring $|D_{b,s}| := n_{b,s} \geq \lfloor n_s/3 \rfloor$ for $b \in [3]$. Using $n., D_{1,s}, D_{2,s}$, and $D_{3,s}$, we estimate $\hat{p}_s, \widehat{\|\beta_s\|}, \tilde{\beta}_s$, and $\hat{\mu}_s$, respectively. The combination of $\hat{p}_s$ and $\widehat{\|\beta_s\|}$ yields $\widehat{\|\beta.\|}$. Furthermore, we generate a duplicate of $D_s$, denoted as $D'_s$, and partition it into $D'_{1,s}$ and $D'_{2,s}$, satisfying $|D'_{b,s}| := n'_{b,s} \geq \lfloor n_s/2 \rfloor$ for $b \in [2]$. We then use $D'_{1,s}$ and $D'_{2,s}$ to estimate $\hat{\beta}'_s$ and $\hat{\mu}'_s$. Precise definitions of the estimator construction and subset partitioning can be found in the appendices.

Incorporating the derived estimators, we construct the final regressor as:

$$\hat{f}_n(x, s) = \widehat{\|\beta.\|}\left\langle \tilde{\beta}_s, x - \hat{\mu}_s \right\rangle + \sum_{s' \in [M]} \hat{p}_{s'}\left\langle \hat{\beta}'_s, \hat{\mu}'_s \right\rangle. \tag{6}$$

# 6 Upper Bound Analyses

In this section, we demonstrate the achievability of the upper bound presented in Theorem 1 utilizing the estimator delineated in Section 5. Initially, we conduct an analysis of the estimator's fairness guarantee, subsequently progressing to an examination of the estimator's mean squared deviation.

## 6.1 Analysis of Fairness

For our fairness guarantee on $\hat{f}_n$, we demonstrate the following theorem.

**Theorem 2.** *If $n \geq 48 \ln(M/\delta)/\min_s p_s$, we have for $\delta \in (0, 1)$,*

$$\mathbb{P}\left\{ \max_{s,s' \in [M]} W_2\left(\nu_{\hat{f}_n|s}, \nu_{\hat{f}_n|s'}\right) > 4B\sigma_X\sigma_X\sqrt{\frac{48\ln(M/\delta)}{\min_{s'' \in [M]} n p_{s''}}} \right\} \leq \delta.$$

By proving Theorem 2, we can immediately confirm that the estimator adheres to $(\alpha, \delta)$-fairness consistency with $\alpha \in (0, 1/2]$.

## 6.2 Analysis of Estimation Error

In this subsection, we derive an upper bound for the estimation error presented in Theorem 1, focusing on the estimator introduced in Section 5. To derive the upper bound in Theorem 1, we begin by decomposing the mean squared deviation of the estimator in Eq (6) as follows:

**Theorem 3.** *For the estimator defined in Eq* (6), *the mean square deviation from* $f_{\mathrm{DP}}^*$ *is bounded above by*

$$
\sum_{s\in[M]} p_s \mathbf{E}\left[\left(\mathbf{E}\left[\widehat{\|\beta_\cdot\|}^2\Big|n_\cdot\right]\right)^{1/2}\mathbf{E}\left[\left\langle\tilde{\beta}_s,\mu_s-\hat{\mu}_s\right\rangle^2\Big|n_\cdot\right]^{1/2}+\sigma_X\mathbf{E}\left[\left(\widehat{\|\beta_\cdot\|}-\overline{\|\beta_\cdot^*\|}\right)^2\Big|n_\cdot\right]^{1/2}+\right.
$$

$$
\sigma_X\overline{\|\beta_\cdot^*\|}\mathbf{E}\left[\left\|\tilde{\beta}_s-\beta_s^*/\|\beta_s^*\|\right\|^2\Big|n_\cdot\right]^{1/2}+\mathbf{E}\left[\left(\sum_{s'\in[M]}\hat{p}_{s'}\left\langle\hat{\beta}'_{s'}-\beta_{s'}^*,\hat{\mu}'_{s'}\right\rangle\right)^2\Big|n_\cdot\right]^{1/2}+
$$

$$
\mathbf{E}\left[\left(\sum_{s'\in[M]}\hat{p}_{s'}\langle\beta_{s'}^*,\hat{\mu}'_{s'}-\mu_{s'}\rangle\right)^2\Big|n_\cdot\right]^{1/2}+\left|\sum_{s'\in[M]}(\hat{p}_{s'}-p_{s'})\langle\beta_{s'}^*,\mu_{s'}\rangle\right|\right)^2\right]. \quad (7)
$$

In Eq (7), the terms correspond to the estimation errors of $\hat{\mu}_s$, $\widehat{\|\beta_\cdot\|}$, $\tilde{\beta}_s$, $\hat{\beta}'_s$, $\hat{\mu}'_s$, and $\hat{p}_s$, respectively. Standard techniques for the OLS estimator and empirical average yield upper bounds for the first, fourth, fifth, and sixth terms. Nevertheless, the second and third terms in Eq (7) involve non-linear transformations of the OLS estimator (i.e., taking the norm or dividing by the norm), complicating their error analysis. This section's primary technical contributions involve establishing tight upper bounds for the second and third terms in Eq (7).

**Estimation error of norm and direction of $\beta_s^*$.** Consider $X_1, ..., X_n \overset{\mathrm{iid}}{\sim} N(\mu, \sigma_X^2 I)$, $\beta^* \in \mathbb{R}^d$ with $\|\beta^*\| \leq B$ for some $B > 0$, and $\xi_1, ..., \xi_n \overset{\mathrm{iid}}{\sim} N(0, \sigma_\xi^2)$. Define $Y_i = \langle\beta^*, X_i\rangle + \xi_i$. The OLS estimator of $\beta^*$ is given by $\hat{\beta} = (\frac{1}{n}X^\top X)^{-1}(\frac{1}{n}X^\top Y)$, where $X = (X_1 \cdots X_n)^\top$ and $Y = (Y_1 \cdots Y_n)^\top$. The direction estimator is $\hat{\beta}/\|\hat{\beta}\|$, while the norm estimator is $\|\hat{\beta}\|$.

We present the estimation errors for direction and norm in Theorems 4 and 5:

**Theorem 4.** *For $n > 6d$, we have*

$$
\mathbf{E}\left[\left\|\frac{\hat{\beta}}{\|\hat{\beta}\|}-\frac{\beta^*}{\|\beta^*\|}\right\|^2\right] \leq \frac{84e^{10}\sigma_\xi^2 d}{\sigma_X^2\|\beta^*\|^2 n}\left(1+\frac{6}{n-6}\right).
$$

**Theorem 5.** *For $n > 6d$, we have*

$$
\mathbf{E}\left[\left(\|\hat{\beta}\|-\|\beta^*\|\right)^2\right] \leq \frac{21e^{10}\sigma_\xi^2 d}{\sigma_X^2 n}\left(1+\frac{6}{n-6}\right).
$$

The direction's estimation error (Theorem 4) is $O(\sigma_\xi^2 d/\sigma_X^2\|\beta^*\|^2 n)$, while the norm's estimation error (Theorem 5) is $O(B^2\sigma_\xi^2 d/\sigma_X^2 n)$. Integrating Theorems 3 to 5 yields the $\sigma_\xi^2 B^2 dM/n$ term in the upper bound in Theorem 1. The remaining part, $U\sigma_\xi^2/n$, arises from the estimation error of $\hat{\beta}'_s$ (the third term in Eq (7)), dominating other terms in Eq (7).

## 7 Lower Bound Analyses

In this section, we provide a proof sketch for the lower bound, outlined in Theorem 1. To facilitate a clear and concise presentation of the proof sketch, we introduce several notations. Let $\theta$ denote the tuple of distribution parameters $(\beta_\cdot, \mu_\cdot)$, and let $\Theta$ represent the set of all such parameters, defined as $\Theta = \mathcal{B} \times \mathcal{M}$. We use $\mathbb{P}_\theta$ and $\mathbf{E}_\theta$ to denote the probability and expectation operators, respectively, given $X \sim N(\mu_S, \sigma_X^2 I)$ and $Y = (\beta_S, X) + \xi$, where $\xi \sim N(0, \sigma_\xi^2)$. We adopt the shorthand $\mathcal{E}(f;\theta) = \mathcal{E}(f;\beta_\cdot,\mu_\cdot)$ for $\theta = (\beta_\cdot,\mu_\cdot)$. Moreover, we define $f_\theta = \arg\min_{f\in\mathcal{F}_{\mathrm{DP}}} R(f;\beta_\cdot,\mu_\cdot)$ for $\theta = (\beta_\cdot,\mu_\cdot)$. For two probability distributions $\pi$ and $\pi'$, the Kullback-Leibler (KL) divergence is denoted as $D_{\mathrm{KL}}(\pi,\pi') = \int \ln(\frac{d\pi}{d\pi'}(z))\pi(dz)$. Finally, we denote the set of all $L^2$ integrable functions $f : \mathbb{R}^d \times [M] \to \mathbb{R}$ as $\mathcal{L}^2$.

By utilizing Fano's inequality, we establish a lower bound for the minimax error as presented in Theorem 1. Due to the invariance of the distribution of $S_1, ..., S_n$ under parameter alterations $\theta$,

Fano's inequality can be applied after conditioning on $S_1, ..., S_n$, or equivalently, $n.$. Consequently, we derive the following theorem:

**Theorem 6.** *Let $\hat{\Theta} \subseteq \Theta$ be a finite set of the parameters such that there exists $\epsilon > 0$ such that for any $\theta, \theta' \in \hat{\Theta}$, $\inf_f \mathcal{E}(f; \theta) \vee \mathcal{E}(f; \theta') \geq \epsilon$, where $\hat{\Theta}$ and $\epsilon$ is possibly dependent on $n.$. Let $|\hat{\Theta}| = K$. Then, for arbitrary $\alpha > 0$ and $\delta \in (0, 1)$, we have*

$$\mathcal{E}_n(\alpha, \delta) \geq \mathbf{E}\left[\epsilon\left(1 - \frac{\inf_\pi \frac{1}{K} \sum_{\theta \in \hat{\Theta}} D_{\mathrm{KL}}\left(\pi_{\theta|n.}, \pi\right) + \ln(2)}{\ln(K)}\right)\right],$$

*where $\pi_{\theta|n.}$ denotes the distribution of $D_n$ conditioned on $n.$ with parameter $\theta$, and the expectation is taken over $n.$.*

As demonstrated in Theorem 6, the lower bound for the minimax error can be obtained by constructing $\hat{\Theta}$ such that: 1) $\inf_f \mathcal{E}(f; \theta) \vee \mathcal{E}(f; \theta') \geq \epsilon$ for any $\theta, \theta' \in \hat{\Theta}$, and 2) $\inf_\pi \frac{1}{K} \sum_{\theta \in \hat{\Theta}} D_{\mathrm{KL}}\left(\pi_{\theta|n.}, \pi\right) \leq \ln(K/4)/2$. With the construction of such a $\hat{\Theta}$, a lower bound of $\mathbf{E}[\frac{\epsilon}{2}]$ is attained.

We present a theorem that establishes a tight lower bound on $\inf_f \mathcal{E}(f; \theta) \vee \mathcal{E}(f; \theta')$.

**Theorem 7.** *Let $\theta$ and $\theta'$ be the parameters of the distributions such that $\frac{1}{2\sigma_X^2}\|\mu_s - \mu_s'\|^2 := d_s < 1$ for all $s \in [M]$. Then, we have*

$$\inf_{f \in \mathcal{L}^2} \mathcal{E}(f; \theta) \vee \mathcal{E}(f; \theta') \geq \sum_{s \in [M]} p_s \frac{\sigma_X^2 e^{-d_s}}{4} \left\| \frac{\overline{\|\beta.\|}\beta_s}{\|\beta_s\|} - \frac{\overline{\|\beta'.\|}\beta_s'}{\|\beta_s'\|} \right\|^2 \left(1 + \frac{d_s}{2}\right)^{1 + \frac{d}{2}}.$$

The term $\|\overline{\|\beta.\|}\beta_s/\|\beta_s\| - \overline{\|\beta'.\|}\beta_s'/\|\beta_s'\|\|^2$ characterizes the lower bound, which is different from the characteristic term in standard linear regression, $\|\beta_s - \beta_s'\|^2$.

We next present the construction of $\hat{\Theta}$. We construct $\hat{\Theta}$ such that each of its elements corresponds to an index from the set $\mathcal{V} = \{-1, 1\}^{M \times (d-1)}$, denoted by $\theta_v = \{\beta_{v,\cdot}, \mu_{v,\cdot}\} \in \hat{\Theta}$, where $\beta_{v,s}$ is controlled such that its norm is equivalent to a specified value $B_s$, i.e., $\|\beta_{v,s}\| = B_s$. This construction ensures that $\hat{\Theta} \subset \mathcal{B} \times \mathcal{M}$. Given positive values $\epsilon_1, ..., \epsilon_M$ and $B_1, ..., B_M$, we construct $\hat{\Theta}$ as follows:

$$\mu_{v,s} = 0, \|\beta_{v,s}\| = B_s, \frac{\beta_{v,s,1}}{\|\beta_{v,s}\|} = \sqrt{1 - \epsilon_s^2}, \text{ and } \frac{\beta_{v,s,i}}{\|\beta_{v,s}\|} = v_{s,i-1}\frac{\epsilon_s}{\sqrt{d-1}} \text{ for } i = 2, ..., d. \quad (8)$$

We demonstrate the following properties for $\hat{\Theta}$ defined in Eq (8).

**Theorem 8.** *Given $\epsilon_1, ..., \epsilon_M > 0$ and $B_1, ..., B_M > 0$, let $\hat{\Theta} \subset \Theta$ represent the set of parameters defined in Eq (8). Let $\pi_{\theta|n.}$ be the distribution of the sample $D_n$ conditioned on $n.$ with the distribution parameter $\theta$. Then, we have 1) for any $v, v' \in \mathcal{V}$,*

$$\inf_{f \in \mathcal{L}^2} \mathcal{E}(f; \theta_v) \vee \mathcal{E}(f; \theta_{v'}) \geq \sum_{s \in [M]} p_s \left(\sum_{s' \in [M]} p_{s'} B_{s'}\right)^2 \frac{\sigma_X^2 \epsilon_s^2}{d-1} d_H(v_s, v's),$$

*and 2) for $v, v' \in \mathcal{V}$,*

$$D_{\mathrm{KL}}\left(\pi_{\theta_v|n.}, \pi_{\theta_{v'}|n.}\right) = \sum_{s \in [M]} \frac{2\sigma_X^2 B_s^2 n_s \epsilon_s^2}{\sigma_\xi^2(d-1)} d_H(v_s, v_s').$$

By integrating Theorems 6 to 8 and employing the renowned Varshamov-Gilbert bound, we derive the lower bound in Theorem 1.

# 8 Conclusion

This paper investigates a regression problem with $(\alpha, \delta)$-fairness consistency as a fairness constraint. Specifically, we demonstrate that, under the constraint of $(\alpha, \delta)$-fairness, the minimax optimal error

scales as $\sigma_\xi^2 B^2 dM/n$ up to a constant factor, when $\alpha \in (0, 1/2]$. Additionally, we provide the fair regressor that achieves this optimal error.

**Potential negative societal impacts.** Our study aims to mitigate the negative impact of regression models on social groups, rather than to cause harm. However, our results are only valid for linear models, as defined in Eq (2). Misapplication of our findings to other models may result in discriminatory treatment, which should be avoided.

## Acknowledgments and Disclosure of Funding

This work was partly supported by JSPS KAKENHI Grant Numbers JP23K13011 and JP23H00483.

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

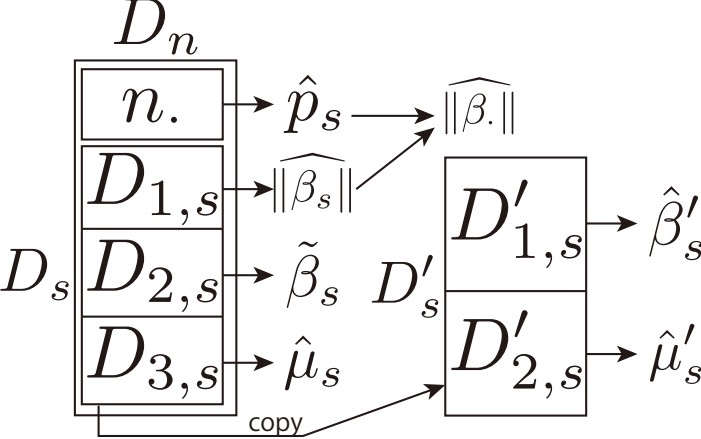

Figure 1: Sample splitting for constructing estimators.

## A    Comparison of Existing and Our Unfairness Scores

This section compares our unfairness score with existing ones. Recall that our unfairness score is defined as the maximum Wasserstein distance between any two distributions $\nu_{f|s}$ and $\nu_{f|s'}$ over all pairs of groups $s$ and $s'$, as follows:

$$U(f) = \max_{s,s'\in[M]} W_2(\nu_{f_n|s}, \nu_{f|s'}).$$

In contrast, Agarwal et al. [2], Chzhen et al. [7, 8] use the Kolmogorov distance $D_{\text{Kol}}$ to measure unfairness, which is defined as:

$$U_{\text{Kol}}(f) = \max_{s,s'\in[M]} D_{\text{Kol}}(\nu_{f|s}, \nu_{f|s'}).$$

The difference between our score and $U_{\text{Kol}}(f)$ is solely the choice of distance metric. Our score utilizes the Wasserstein distance, while $U_{\text{Kol}}(f)$ uses the Kolmogorov distance. This difference arises mainly from technical reasons.

In addition, Chzhen and Schreuder [6] proposed another unfairness score, denoted by $U_{\text{AvgW}_2}(f)$, which is defined as the average of the Wasserstein distance, as follows:

$$U_{\text{AvgW}_2}(f) = \inf_\nu \sum_{s\in[M]} p_s W_2\big(\nu_{f|s}, \nu\big).$$

Here, the score places more emphasis on the major groups, as reflected by the weight of $p_s$. This may not be desirable if the unfairness is more prevalent in the minority groups, which may be common in real-world scenarios.

## B    Estimator Details

This section describes the construction of our optimal estimator in detail. Recall that our estimator is a plugin estimator in which we first estimate the parts of the terms in Eq (5) and then substitute them into Eq (5). Specifically, we construct estimators for $\widehat{\|\beta.\|}$, $\tilde{\beta}_s$, $\hat{p}_{s'}$, $\hat{\beta}'_{s'}$, and $\hat{\mu}'_{s'}$, where they correspond to the terms in Eq (5) as follows:

$$\underbrace{\overline{\|\beta^*_.\|}}_{\widehat{\|\beta.\|}}\Big\langle \underbrace{\frac{\beta^*_s}{\|\beta^*_s\|}, x - \underbrace{\mu_s}_{\hat{\mu}_s}}_{\tilde{\beta}_s}\Big\rangle + \sum_{s'\in[M]} \underbrace{p_{s'}}_{\hat{p}_{s'}}\Big\langle \underbrace{\beta^*_{s'}}_{\hat{\beta}'_{s'}}, \underbrace{\mu_{s'}}_{\hat{\mu}'_{s'}}\Big\rangle.$$

**Algorithm 1:** Algorithm of the proposed optimal estimator.
***
**Input** : The sample $D_n = \{(X_i, S_i, Y_i)\}_{i=1}^{n}$.
**Output** : The regressor $\hat{f}_n$.
**for** $s \leftarrow 1$ **to** $M$ **do**

    Calculate $n_s$ and construct the group-wise sample $D_s$ and its duplicate $D'_s$ ;
    Partition $D_s$ into equal-sized subsets: $D_{1,s}$, $D_{2,s}$, and $D_{3,s}$ ;

    Compute the estimands $\hat{p}_s$ from $n.$, $\widehat{\|\beta_s\|}$ from $D_{1,s}$, $\tilde{\beta}_s$ from $D_{2,s}$, and $\hat{\mu}_s$ from $D_{3,s}$ ;
    Partition $D'_s$ equally into $D'_{1,s}$ and $D'_{2,s}$ ;

    Compute the estimands $\hat{\beta}'_s$ from $D'_{1,s}$ and $\hat{\mu}'_s$ from $D'_{2,s}$ ;

**end**
Calculate $\hat{f}_n$ using Eq (6) ;
**return** $\hat{f}_n$
***

For analysis purposes, we split the sample into several subsets and pass each subset to the corresponding estimator (the correspondence is explained later). Figure 1 shows an overview of the sample splitting and the correspondence between the subsets and estimators. We construct the histogram of the sensitive attribute $S_i$ from the sample $D_n$, denoted as $n. = (n_1, ..., n_M)$, where $n_s = |\{i \in [n] : S_i = s\}|$ (upper left in Figure 1). We also construct group-wise samples $D_s = \{(X_i, Y_i) : i \in [n], S_i = s\}$. For each $s \in [M]$, we divide $D_s$ into $D_{1,s}$, $D_{2,s}$, and $D_{3,s}$ such that $|D_{b,s}| := n_{b,s} \geq \lfloor n_s/3 \rfloor$ for $b \in [3]$ (lower left in Figure 1). We use $n.$, $D_{1,s}$, $D_{2,s}$, and $D_{3,s}$ to estimate $\hat{p}s$, $\widehat{\|\beta_s\|}$, $\tilde{\beta}_s$, and $\hat{\mu}_s$, respectively, where $\widehat{\|\beta_s\|}$ is the estimator for $\|\beta^*s\|$. We obtain $\widehat{\|\beta.\|}$ from the combination of $\hat{p}_s$ and $\widehat{\|\beta_s\|}$ (middle in Figure 1). Furthermore, for each $s \in [M]$, we create a copy of $D_s$, denoted as $D's$, and divide it into $D'1, s$ and $D'2, s$ such that $|D'b, s| := n'b, s \geq \lfloor n_s/2 \rfloor$ for $b \in [2]$. We use $D'1, s$ and $D'_{2,s}$ to estimate $\hat{\beta}'_s$ and $\hat{\mu}'_s$, respectively (right in Figure 1).

We describe the construction of each estimator below. We first define some notations. Let the $i$th element of $D_{b,s}$ and $D'_{b,s}$ be denoted as $(X_{b,s,i}, Y_{b,s,i})$ and $(X'_{b,s,i}, Y'_{b,s,i})$, respectively. We use the matrix notations $X_{b,s} = (X_{b,s,1} \cdots X_{b,s,n_{b,s}})^\top$, $X'_{b,s} = (X'_{b,s,1} \cdots X'_{b,s,n'_{b,s}})^\top$, $Y_{b,s} = (Y_{b,s,1} \cdots Y_{b,s,n_{b,s}})^\top$, and $Y'_{b,s} = (Y'_{b,s,1} \cdots Y'_{b,s,n'_{b,s}})^\top$. We define the ordinary least square estimators for the subset of the sample $D_{b,s}$ and $D'_{b,s}$ as follows:

$$\hat{\beta}_{b,s} = \left(\frac{1}{n_{b,s}} X_{b,s}^\top X_{b,s}\right)^{-1} \left(\frac{1}{n_{b,s}} X_{b,s}^\top Y_{b,s}\right),$$

$$\hat{\beta}'_{b,s} = \left(\frac{1}{n'_{b,s}} (X'_{b,s})^\top X'_{b,s}\right)^{-1} \left(\frac{1}{n'_{b,s}} (X'_{b,s})^\top Y'_{b,s}\right).$$

We construct each estimator as follows:

($\hat{p}s$)    We use the empirical mean defined as $\hat{p}_s = n_s/n$.

($\widehat{\|\beta_s\|}$)    We use the norm of the OLS estimator. We define $\widehat{\|\beta_s\|} = \|\hat{\beta}_{1,s}\|$ if $n_s > 18d$, and $\widehat{\|\beta_s\|} = 0$ otherwise.

($\widehat{\|\beta.\|}$)    Since $\|\beta^*\| = \sum_{s \in [M]} p_s \|\beta_s^*\|$, we construct its estimator as $\widehat{\|\beta.\|} = \sum_{s \in [M]} \hat{p}_s \widehat{\|\beta_s\|}$.

($\tilde{\beta}_s$)    We use the normalized ordinary least square estimator; $\tilde{\beta}_s = \hat{\beta}_{2,s}/\|\hat{\beta}_{2,s}\|$ if $n_s > 18d$, and $\tilde{\beta}_s = 0$ otherwise.

($\hat{\mu}_s$)    We use the empirical mean; $\hat{\mu}_s = \frac{1}{n_{3,s}} \sum_{i=1}^{n_{3,s}} X_{3,s,i}$.

($\hat{\beta}'_s$)    We employ the ordinary least square estimator; $\hat{\beta}'_s = \hat{\beta}'_{1,s}$ if $n_s > 12d$, and $\hat{\beta}'_s = 0$ otherwise.

($\hat{\mu}'_s$)    We use the empirical mean; $\hat{\mu}'_s = \frac{1}{n'_{2,s}} \sum_{i=1}^{n'_{2,s}} X'_{2,s,i}$ if $n_s > 12d$, and $\hat{\mu}'_s = 0$ otherwise.

Some estimators change their behavior based on the condition $n_s > 18d$ or $n_s > 12d$, which is done for technical purposes in later analyses.

Recall that the final regressor is constructed as follows:

$$\hat{f}_n(x,s) = \widehat{\|\beta.\|}\left\langle \tilde{\beta}_s, x - \hat{\mu}_s \right\rangle + \sum_{s'\in[M]} \hat{p}_{s'}\left\langle \hat{\beta}'_s, \hat{\mu}'_s \right\rangle.$$

Algorithm 1 shows the algorithm for our estimator.

## C Bayes Optimal Regressor under Our Modell

This section presents the proof of Lemma 1, demonstrating the Bayes optimal regressor under the model Eq (2). To establish this, we make use of a key result from the work of Chzhen et al. [7]:

**Theorem 9** (Chzhen et al. [7]). *Assume, for each $s \in [M]$, $\nu_{f^*|s}$ has a density. Then,*

$$\inf_{f:\mathrm{DP}} \mathbf{E}\Big[(f(X,S) - f^*(X,S))^2\Big] = \inf_{\nu} \sum_{s\in[M]} p_s W_2^2(\nu_{f^*|s}, \nu).$$

*where the infimum is taken over all the regressors that satisfy the demographic parity. Moreover, letting $f^*_{\mathrm{DP}}$ and $\nu^*$ be the minimizer of the lhs and rhs, respectively, we have $\nu_{f^*_{\mathrm{DP}}} = \nu^*$ and*

$$f^*_{\mathrm{DP}}(x,s) = \left( \sum_{s'\in[M]} p_{s'} F^{-1}_{f^*|s'} \right) \circ F_{f^*|s}(f^*(x,s)). \tag{9}$$

Here, we denote $\nu_{f^*}$ as the distribution of $f^*(X,S)$, $F_{f|s}$ as the cumulative distribution function of $f(X,S)$ conditioned on $S = s$, and $F^{-1}_{f|s}$ as the inverse cumulative distribution function, given by $F^{-1}_{f|s}(t) = \inf\{y \in \mathbb{R} | F_{f|s}(y) \geq t\}$.

Building upon the results of Theorem 9, we establish the proof of Lemma 1.

*Proof of Lemma 1.* Building upon Theorem 9, we can derive the Bayes optimal regressor under the model Eq (2) by obtaining closed expressions of the cumulative and inverse cumulative distribution functions $F_{f^*|s}$ and $F^{-1}_{f^*|s}$. To obtain these closed forms, we apply certain transformations to $f^*(X,S)$ that render it a random variable following a standard normal distribution. Let $\Phi$ and $\Phi^{-1}$ be the CDF and inverse CDF of the standard normal distribution, respectively. Through elementary calculations, we have:

$$
\begin{aligned}
F_{f^*|s}(t) =& \mathbb{P}\{f^*(X,S) \leq t | S = s\}\\
=& \mathbb{P}\{\langle \beta^*_s, X \rangle \leq t | S = s\}\\
=& \mathbb{P}\left\{ \frac{1}{\sigma_X \|\beta^*_s\|}\langle \beta^*_s, X - \mu_s \rangle \leq \frac{1}{\sigma_X \|\beta^*_s\|}(t - \langle \beta^*_s, \mu_s \rangle) | S = s \right\}.
\end{aligned}
$$

Here, we can readily observe that $\frac{1}{\sigma_X \|\beta^*_s\|}\langle \beta^*_s, X - \mu_s \rangle$ follows the standard normal distribution under conditioned on $S = s$, as $X \sim N(\mu_s, \sigma_X I)$ conditioned on $S = s$. Consequently, we have

$$F_{f^*|s}(t) = \Phi\left( \frac{1}{\sigma_X \|\beta^*_s\|}(t - \langle \beta^*_s, \mu_s \rangle) \right). \tag{10}$$

The inverse function of $F^{-1}_{f^*|s}(t)$ can be obtained by equating the right-hand side to $p$ and solving the resulting equation for $t$, which leads to

$$F^{-1}_{f^*|s}(p) = \sigma_X \|\beta^*_s\| \Phi^{-1}(p) + \langle \beta^*_s, \mu_s \rangle. \tag{11}$$

By substituting Eqs (10) and (11) into Eq (9) in Theorem 9, we obtain the desired claim. $\square$

## D Details of Fairness Analysis

In this section, we provide evidence of the guarantee of our estimator's fairness consistency. Specifically, we present the following theorem.

**Theorem 10.** *For any $\delta \in (0,1]$, the regressor in Eq (6) is $(1/2, \delta)$-consistently fair.*

We prove the above claim by utilizing Theorem 2, which is shown in the main body, as follows:

*Proof of Theorem 10.* We can confirm the claim by comparing the bound obtained in Theorem 2 with the definition of $(\frac{1}{2}, \delta)$-consistent fairness in Definition 2. In particular, we can set $C = 4B\sigma_X \sqrt{\frac{48 \ln(M/\delta)}{\min_{s \in [M]} p_s}}$ and $n_0 = n \geq 48 \ln(M/\delta)/\min_s p_s$ to satisfy the definition of $(\frac{1}{2}, \delta)$-consistent fairness. $\qquad \square$

Next, we provide the proof of Theorem 2. To this end, we prove the following two theorems:

**Theorem 11.** *Let $\hat{f}_n$ be the estimator of $f_{\mathrm{DP}}^*$ defined in Eq (6). Then, almost surely, we have*

$$W_2\left(\nu_{\hat{f}_n|s}, \nu_{\hat{f}_{fn}|s'}\right) \leq 2B\left(\left|\left\langle \tilde{\beta}_s, \mu_s - \hat{\mu}_s \right\rangle\right| \vee \left|\left\langle \tilde{\beta}_{s'}, \mu_{s'} - \hat{\mu}_{s'} \right\rangle\right|\right).$$

**Theorem 12.** *If $n \geq (48 \ln(M/\delta) - 36d)/\min_s p_s$, we have for $\delta \in (0,1)$,*

$$\mathbb{P}\left\{\exists s \in [M], \left|\left\langle \tilde{\beta}_s, \mu_s - \hat{\mu}_s \right\rangle\right| > \sigma_X \sqrt{\frac{48 \ln(M/\delta)}{\min_s np_s + 36d}}\right\} \leq \delta.$$

Combining Theorems 11 and 12 immediately yields Theorem 2.

*Proof of Theorem 11.* This proof investigates the distribution of $\nu_{\hat{f}n|s}$. It is straightforward to verify that, conditioned on $S = s$, $\hat{f}_n(X, S)$ follows the Gaussian distribution with mean

$$\widehat{\|\beta.\|}\left\langle \tilde{\beta}_s, \mu_s - \hat{\mu}_s \right\rangle + \sum_{s' \in [M]} \hat{p}_{s'} \left\langle \hat{\beta}_{s'}', \hat{\mu}_{s'}' \right\rangle,$$

and variance

$$\sigma_X^2 \widehat{\|\beta.\|}^2.$$

We can thus evaluate the Wasserstein distance between the distributions $\nu_{\hat{f}n|s}$ and $\nu \hat{f}n|s'$ using the Wasserstein distance between Gaussian distributions. Given two Gaussian distributions $N(\mu, \sigma^2)$ and $N(\mu', \sigma'^2)$, the 2-Wasserstein distance between them are obtained [20] as

$$W_2^2(N(\mu, \sigma^2), N(\mu', \sigma^2)) = (\mu - \mu')^2 + (\sigma - \sigma')^2.$$

Therefore, we have

$$W_2^2(\nu_{\hat{f}_n|s}, \nu_{\hat{f}_n|s'}) = \widehat{\|\beta.\|}^2 \left(\left\langle \tilde{\beta}_s, \mu_s - \hat{\mu}_s \right\rangle - \left\langle \tilde{\beta}_{s'}, \mu_{s'} - \hat{\mu}_{s'} \right\rangle\right)^2$$

$$\leq 4B^2\left(\left|\left\langle \tilde{\beta}_s, \mu - \hat{\mu} \right\rangle\right| \vee \left|\left\langle \tilde{\beta}_{s'}, \mu - \hat{\mu} \right\rangle\right|\right)^2,$$

which concludes the claim. $\qquad \square$

*Proof of Theorem 12.* We start by deriving the concentration inequality for $\langle \tilde{\beta}_s, \mu_s - \hat{\mu}_s \rangle$ conditioned on $\tilde{\beta}_s$ and $n.$. Note that $\tilde{\beta}_s = 0$ if $n_s \leq 18d$. Conditioning on $\tilde{\beta}_s$ and $n.$, we observe that $\langle \tilde{\beta}_s, \mu_s - \hat{\mu}_s \rangle$ follows a Gaussian distribution with mean zero and variance $\sigma_X^2/n_{3,s}$. Therefore, for any $s \in [M]$ and $t > 0$,

$$\mathbb{P}\left\{\left\langle \tilde{\beta}_s, \mu_s - \hat{\mu}_s \right\rangle > t \Big| \tilde{\beta}_s, n.\right\} \leq \mathbb{1}\{n_s > 18d\} \exp\left(-\frac{n_{3,s}t^2}{2\sigma_X^2}\right).$$

Taking the expectation with respect to $\tilde{\beta}_s$ and using the fact that $n_{3,s} \geq \lfloor n_s/3 \rfloor \geq n_s/6$ for $n_s \geq 6$, we obtain the following inequality for $s \in [M]$ and $t > 0$:

$$\mathbb{P}\left\{\left\langle \tilde{\beta}_s, \mu_s - \hat{\mu}_s \right\rangle > t \Big| n.\right\} \leq \mathbb{1}\{n_s > 18d\} \exp\left(-\frac{n_s t^2}{12\sigma_X^2}\right).$$

Using the union bound for $t > 0$, we have

$$\mathbb{P}\left\{\exists s \in [M], \left\langle \tilde{\beta}_s, \mu_s - \hat{\mu}_s \right\rangle > t \,\middle|\, n. \right\} \le \sum_{s \in [M]} \mathbb{1}\{n_s > 18d\} \exp\left(-\frac{n_s t^2}{12\sigma_X^2}\right). \tag{12}$$

We now derive a sufficient condition on $t$ such that the expectation of the right-hand side in Eq (12) is less than $\delta$. First, we note that

$$\mathbb{1}\{n_s > 18d\} \exp\left(-\frac{n_s t^2}{12\sigma_X^2}\right)$$

$$\le \mathbb{1}\{n_s > 18d\} \exp\left(-\frac{(n_s + 18d)t^2}{12\sigma_X^2} \frac{n_s}{n_s + 18d}\right)$$

$$\le \exp\left(-\frac{(n_s + 18d)t^2}{24\sigma_X^2}\right)$$

Taking the expectation and substituting the moment-generating function of the binomial distribution, we obtain

$$\mathbf{E}\left[\exp\left(-\frac{(n_s + 18d)t^2}{24\sigma_X^2}\right)\right]$$

$$\le \exp\left(-\frac{18dt^2}{24\sigma_X^2}\right)\left(1 - p_s + p_s e^{-\frac{t^2}{24\sigma_X^2}}\right)^n$$

$$\le \exp\left(-\frac{18dt^2}{24\sigma_X^2} - np_s\left(1 - e^{-\frac{t^2}{24\sigma_X^2}}\right)\right).$$

Since $1 - e^{-x} \ge (1 - e^{-1})x$ for $x \in [0, 1]$, if $t^2/24\sigma_X^2 \le 1$, we have

$$\mathbf{E}\left[\exp\left(-\frac{(n_s + 18d)t^2}{24\sigma_X^2}\right)\right]$$

$$\le \exp\left(-\frac{18dt^2}{24\sigma_X^2} - \frac{(1 - e^{-1})np_s t^2}{24\sigma_X^2}\right).$$

Hence, $\mathbf{E}[\exp(-\frac{(n_s+18d)t^2}{24\sigma_X^2})] \le \delta/M$ if $t \ge \sigma_X\sqrt{\frac{24\ln(M/\delta)}{(1-e^{-1})np_s+18d}} \le \sigma_X\sqrt{\frac{48\ln(M/\delta)}{\min_s np_s+36d}}$ because $(1 - e^{-1}) \ge 1/2$. To ensure $t^2/24\sigma_X^2 \le 1$, we require $n \ge (48\ln(M/\delta) - 36d)/\min_s p_s$. $\qquad\square$

## E   Proofs for Norm and Direction Estimators

This section presents the proofs for Theorem 4 and Theorem 5. Our strategy for proving these theorems is to use the hyperellipsoid to interpret the distribution of the OLS estimator. Specifically, we begin by defining $\Sigma_n = \frac{1}{n}X^\top X$ and expressing the OLS estimator $\hat{\beta}$ as

$$\hat{\beta} = \beta^* + \Sigma_n^{-1}\left(\frac{1}{n}X^\top \xi\right), \tag{13}$$

where $\xi$ follows a zero-mean Gaussian distribution. Eq (13) shows that, conditioned on $X$, $\hat{\beta}$ follows a multivariate Gaussian distribution with mean $\beta^*$ and covariance matrix $\frac{\sigma_\xi^2}{n}\Sigma_n^{-1}$. We establish that, under the condition $\|\hat{\beta}\| = r$, $\hat{\beta}$ is supported on a hyperellipsoid $E(r, \beta \cdot \frac{n}{\sigma_\xi^2}\Sigma_n)$, where $E(r, c, A) = \{x \in \mathbb{R}^d : (x - c)^\top A(x - c) \le r\}$ denotes the hyperellipsoid with $r > 0$, $c \in \mathbb{R}^d$, and a symmetric and positive-definite matrix $A \in \mathbb{R}^{d \times d}$.

To prove Theorem 4 and Theorem 5, we adopt the following strategy. First, we provide an approximation of the hyperellipsoid $E(r, c, A)$ using the maximum eigenvalue of $A^{-1}$, i.e., $\lambda_{\max}(A^{-1})$. In our context, $A = \frac{n}{\sigma_\xi^2}\Sigma_n$, and we then focus on the concentration inequalities regarding $\lambda_{\max}(\Sigma_n^{-1})$. Finally, we combine these tools to prove both theorems.

**Lemmas regarding hyperellipsoid.** We present two lemmas that relate to the approximation of the hyperellipsoid $E(r, c, A)$. Specifically, we demonstrate the following two lemmas:

**Lemma 2.** *For $r > 0$, $c \in \mathbb{R}^d$, and a symmetric and positive-definite matrix $A \in \mathbb{R}^{d \times d}$, we have $E(r, c, A) \subseteq E(r\lambda_{\max}(A^{-1}), c, I)$.*

**Lemma 3.** *For $r > 0$, $c \in \mathbb{R}^d$, and a symmetric and positive-definite matrix $A \in \mathbb{R}^{d \times d}$, if $r\lambda_{\max}(A^{-1}) \leq \|c\|^2$, we have*

$$\inf_{x \in E(r,c,A)} \left\langle \frac{c}{\|c\|}, \frac{x}{\|x\|} \right\rangle \geq \sqrt{1 - \frac{r}{\|c\|^2} \lambda_{\max}(A^{-1})}.$$

These lemmas provide insight into the approximation of the hyperellipsoid $E(r, c, A)$ for a given positive value of $r$, vector $c$ in $\mathbb{R}^d$, and positive-definite symmetric matrix $A$ in $\mathbb{R}^{d \times d}$. Lemma 2 states that the hyperellipsoid $E(r, c, A)$ is contained within a hyperellipsoid $E(r\lambda_{\max}(A^{-1}), c, I)$. Lemma 3 shows that, under certain conditions, the minimum angle between a point in $E(r, c, A)$ and the vector $c$ is bounded below by a quantity that depends on $r$, $c$, and $A$.

*Proof of Lemma 2.* It is trivial that $A - \lambda_{\min}(A)I$ is positive semi-definite. Equivalently, we have for any $x \in \mathbb{R}^d$,

$$x^\top (A - \lambda_{\min}(A)I)x \geq 0$$
$$\Longleftrightarrow x^\top Ax \geq x^\top \lambda_{\min}(A)Ix. \tag{14}$$

From Eq (14), for any $x \in E(r, c, A)$, we have

$$x^\top \lambda_{\min}(A)Ix \leq x^\top Ax \leq r.$$

Hence, for any $x \in E(r, c, A)$, we have

$$x^\top Ix \leq \frac{r}{\lambda_{\min}(A)} = \lambda_{\max}(A^{-1})r,$$

which indicates $x \in E(r\lambda_{\max}(A^{-1}), c, I)$. $\qquad \square$

*Proof of Lemma 3.* Let $\bar{c} = c/\|c\|$, and define a set $\bar{E}(r, c, A) = \{x \in \mathbb{S}d - 1 : \exists \gamma > 0, \gamma x \in E(r, c, A)\}$. Then, $x \in \bar{E}(r, c, A)$ if and only if

$$\inf_{\gamma > 0} (\gamma x - c)^\top A(\gamma x - c) \leq r. \tag{15}$$

We can rewrite the left-hand side of Eq (15) as

$$\gamma^2 \langle x, Ax \rangle - 2\gamma \langle c, Ax \rangle + \langle c, Ac \rangle$$
$$= \langle x, Ax \rangle \left( \gamma - \frac{\langle c, Ax \rangle}{\langle x, Ax \rangle} \right)^2 + \langle c, Ac \rangle - \frac{\langle x, Ac \rangle^2}{\langle x, Ax \rangle}.$$

Hence,

$$\inf_{\gamma > 0} (\gamma x - c)^\top A(\gamma x - c) = \langle c, Ac \rangle - \frac{(\langle x, Ac \rangle \vee 0)^2}{\langle x, Ax \rangle}.$$

Consequently, $x \in \bar{E}(r, c, A)$ if and only if

$$\langle x, Ax \rangle \left( \langle \bar{c}, A\bar{c} \rangle - \frac{r}{\|c\|^2} \right) \leq (\langle x, A\bar{c} \rangle \vee 0)^2. \tag{16}$$

From Lemma 2, we have

$$\inf_{x \in E(r,c,A)} \left\langle \bar{c}, \frac{x}{\|x\|} \right\rangle \geq \inf_{x \in E(\lambda_{\max}(A^{-1})r,c,I)} \left\langle \bar{c}, \frac{x}{\|x\|} \right\rangle$$
$$= \inf_{x \in \bar{E}(\lambda_{\max}(A^{-1})r,c,I)} \langle \bar{c}, x \rangle. \tag{17}$$

By Eq (16), $x \in \bar{E}(\lambda_{\max}(A^{-1})r, c, I)$ if and only if

$$1 - \frac{r}{\|c\|^2} \lambda_{\max}(A^{-1}) \leq (\langle x, \bar{c} \rangle \vee 0)^2. \tag{18}$$

Combining Eqs (17) and (18) and the assumption yields the claim. $\qquad \square$

**Least eigenvalue of the empirical covariance matrix.** The previous lemmas, Lemmas 2 and 3, provide valuable insight into analyzing the randomness regarding $\xi$. However, to account for the randomness of $X$, we must also control the lower bound on the least eigenvalue of $A$ in Lemmas 2 and 3, which corresponds to the least eigenvalue of $\frac{1}{n}X^\top X$ in our context. To this end, we leverage the high probability bound presented by Mourtada [15] based on the small-ball condition. We state the following probabilistic bound and expectation bound.

**Lemma 4.** *For $\mu \in \mathbb{R}^d$ and $\sigma_X^2 > 0$, let $X_1, ..., X_n \overset{\text{iid}}{\sim} N(\mu, \sigma_X^2 I)$, and let $X = (X_1 \cdots X_n)^\top$. Then, for $n > 6d$, we have*

$$\mathbb{P}\left\{\lambda_{\min}\left(\frac{1}{n}X^\top X\right) < t\right\} \leq \left(\frac{21e^{10}}{\sigma_X^2}t\right)^{n/6}.$$

**Lemma 5.** *For $\mu \in \mathbb{R}^d$ and $\sigma_X^2 > 0$, let $X_1, ..., X_n \overset{\text{iid}}{\sim} N(\mu, \sigma_X^2 I)$, and let $X = (X_1 \cdots X_n)^\top$. Then, for $n > 6d$, we have*

$$\mathbf{E}\left[\lambda_{\max}\left(\left(\frac{1}{n}X^\top X\right)^{-1}\right)\right] \leq \frac{21e^{10}}{\sigma_X^2}\left(1 + \frac{6}{n-6}\right).$$

To prove Lemma 4, we utilize Corollary 3 in Mourtada [15]. Specifically, we use the following theorem.

**Theorem 13** (Corollary 3 in Mourtada [15]). *Let $X$ be a random vector in $\mathbb{R}^d$ such that $\mathbf{E}[\|X\|^2] < +\infty$, and let $\Sigma = \mathbf{E}[XX^\top]$. Let $\hat{\Sigma}_n = \frac{1}{n}\sum_{i=1}^n X_i X_i^\top$, where $X_i$ are i.i.d. copies of $X$. Given $C > 0$ and $\alpha \in (0, 1]$, assume that for every $\theta \in \mathbb{R}^d \setminus \{0\}$ and $t > 0$,*

$$\mathbb{P}\left\{\langle\theta, X\rangle^2 \leq t^2\left\|\Sigma^{1/2}\theta\right\|^2\right\} \leq (Ct)^\alpha. \tag{19}$$

*Then, if $d/n \leq \alpha/6$, for every $t > 0$,*

$$\hat{\Sigma}_n \succeq t\Sigma$$

*with probability at least $1 - (C't)^{\alpha n/6}$, where $C' = 3C^4 e^{1+9/\alpha}$.*

Eq (19) is known as the small-ball condition.

*Proof of Lemma 4.* To take an advantage of Theorem 13, we need to ensure that $X_i$ satisfies the small-ball condition in Eq (19). Let $\Sigma_n = \frac{1}{n}X^\top X$. Then, the expected value of $\Sigma_n$ is equal to $\sigma_X^2 I + \mu\mu^\top := \Sigma$, i.e., $\mathbf{E}[\Sigma_n] = \Sigma$. Given $\theta \in \mathbb{R}^d \setminus \{0\}$, $\langle\theta, X_i\rangle^2/\sigma_X^2\|\theta\|^2$ follows the non-central $\chi^2$ distribution with degree of freedom 1 and non-centrality parameter $\langle\theta, \mu\rangle^2/\sigma_X^2\|\theta\|^2$. Consequently, we verify the satisfication of the small-ball condition of $X_i$ by confirming that for a random variable $Z$ following the non-central $\chi^2$ distribution with degree of freedom 1 and non-centrality parameter $\lambda^2$, there exists $C$ and $\alpha \in (0, 1]$ such that

$$\mathbb{P}\{Z \leq t^2\} \leq (Ct)^\alpha.$$

The cumulative distribution fucntion of the non-central $\chi^2$ distribution with degree of freedom 1 has a closed-form using the error function (See [12] and references therein). Specifically, letting $\text{erf}(z)$ be the error function, defined as

$$\text{erf}(z) = \frac{2}{\sqrt{\pi}}\int_0^z e^{-x^2}dx,$$

the cumulative distribution function of $Z$ is obtained as

$$\mathbb{P}\{Z \leq t^2\} = \frac{1}{2}\left(\text{erf}\left(\frac{t-\lambda}{\sqrt{2}}\right) + \text{erf}\left(\frac{t+\lambda}{\sqrt{2}}\right)\right).$$

Since $e^{-x^2}$ is an even function, we have

$$\mathbb{P}\{Z \leq t^2\} = \frac{1}{\sqrt{2\pi}}\left(\int_0^{\lambda+t} e^{-x^2/2}dx + \int_0^{t-\lambda} e^{-x^2/2}dx\right)$$

$$= \frac{1}{\sqrt{2\pi}} \left( \int_0^{\lambda+t} e^{-x^2/2} dx + \int_{\lambda-t}^0 e^{-x^2/2} dx \right)$$

$$= \frac{1}{\sqrt{2\pi}} \left( \int_0^{\lambda+t} e^{-x^2/2} dx - \int_0^{\lambda-t} e^{-x^2/2} dx \right)$$

$$= \frac{1}{\sqrt{2\pi}} \int_{\lambda-t}^{\lambda+t} e^{-x^2/2} dx.$$

Noting that $e^{-x^2/2} \leq e^{-\frac{(0\vee(\lambda-t))^2}{2}}$ for $x \in (\lambda - t, \lambda + t)$, we have

$$\mathbb{P}\{Z \leq t^2\} \leq \frac{1}{\sqrt{2\pi}} \int_{\lambda-t}^{\lambda+t} dx = \sqrt{\frac{2}{\pi}} e^{-\frac{(0\vee(\lambda-t))^2}{2}} t. \tag{20}$$

We verify that $X_i$ satisfies the small-ball condition by utilizing Eq (20). Recall that $\langle \theta, X_i \rangle^2 / \sigma_X^2 \|\theta\|^2$ follows the non-central $\chi^2$ distribution with degree of freedom 1 and non-centrality parameter $\lambda^2 = \langle \theta, \mu \rangle^2 / \sigma_X^2 \|\theta\|^2$ for any $\theta \in \mathbb{R}^d \setminus \{0\}$. By Eq (20), we have

$$\mathbb{P}\left\{ \frac{|\langle \theta, X_i \rangle|^2}{\sigma_X^2 \|\theta\|^2} < t^2 \right\} \leq \sqrt{\frac{2}{\pi}} e^{-\frac{(0\vee(\lambda-t))^2}{2}} t.$$

Noting that $\left\| \Sigma^{1/2}\theta \right\|^2 = \sigma_X^2 \|\theta\|^2 + \langle \theta, \mu \rangle^2$, we have

$$\mathbb{P}\left\{ \langle \theta, X_i \rangle^2 \leq t^2 \left\| \Sigma^{1/2}\theta \right\|^2 \right\} \leq \sqrt{\frac{2}{\pi} \left( 1 + \frac{\langle \theta, \mu \rangle^2}{\sigma_X^2 \|\theta\|^2} \right)} e^{-\frac{(0\vee(\lambda-t))^2}{2}} t$$

$$= \sqrt{\frac{2}{\pi}(1 + \lambda^2)} e^{-\frac{(0\vee(\lambda-t))^2}{2}} t. \tag{21}$$

We divide into two cases, $\lambda > t$ and $\lambda \leq t$, to derive an upper bound on Eq (21).

(Case $\lambda > t$) Since $(\lambda - t)^2 = \lambda^2 - 2\lambda t + t^2 \geq \lambda^2 - 2t^2 + t^2 = \lambda^2 - t^2$, an upper bound on Eq (21) is obtained as

$$\sqrt{\frac{2}{\pi}(1 + \lambda^2)} e^{-\frac{(0\vee(\lambda-t))^2}{2}} t$$

$$\leq \sqrt{\frac{2}{\pi}(1 + \lambda^2)} e^{-\frac{\lambda^2-t^2}{2}} t.$$

For positive numbers $a$ and $b$, $\sqrt{a + b} \leq \sqrt{a} + \sqrt{b}$. Using this fact, we have

$$\sqrt{\frac{2}{\pi}(1 + \lambda^2)} e^{-\frac{(0\vee(\lambda-t))^2}{2}} t$$

$$\leq \sqrt{\frac{2}{\pi}} \left( \sqrt{1 + t^2} + \sqrt{\lambda^2 - t^2} e^{-\frac{\lambda^2-t^2}{2}} \right) t.$$

Since a function $x \to xe^{-x^2/2}$ admits a maximum on $x \in (0, \infty)$ of $e^{-1/2}$, we have $\sqrt{\lambda^2 - t^2} e^{-\frac{\lambda^2-t^2}{2}} \leq e^{-1/2}$. Consequently, we have

$$\sqrt{\frac{2}{\pi}(1 + \lambda^2)} e^{-\frac{(0\vee(\lambda-t))^2}{2}} t \leq \sqrt{\frac{16}{\pi e}(1 + t^2)} t, \tag{22}$$

where we use the fact $(1 + e^{-1/2})^2 \leq 8e^{-1}$.

(Case $\lambda \leq t$) We can easily verify that

$$\sqrt{\frac{2}{\pi}(1 + \lambda^2)} e^{-\frac{(0\vee(\lambda-t))^2}{2}} t \leq \sqrt{\frac{2}{\pi}(1 + t^2)} t \tag{23}$$

Combining Eqs (22) and (23), we have for every $t > 0$,

$$\mathbb{P}\left\{ \langle \theta, X_i \rangle^2 \leq t^2 \left\| \Sigma^{1/2}\theta \right\|^2 \right\} \leq \sqrt{\frac{16}{\pi e}(1 + t^2)t}.$$

For $t^2 \in (0, -\frac{1}{2} + \frac{1}{2}\sqrt{1 + \frac{\pi e}{4}}]$, we have

$$\sqrt{\frac{16}{\pi e}(1 + t^2)t} \leq \sqrt{\frac{8}{\pi e}\left(1 + \sqrt{1 + \frac{\pi e}{4}}\right)t}.$$

For $t \geq -\frac{1}{2} + \frac{1}{2}\sqrt{1 + \frac{\pi e}{4}}$,

$$\sqrt{\frac{8}{\pi e}\left(1 + \sqrt{1 + \frac{\pi e}{4}}\right)t} \geq \sqrt{\frac{4}{\pi e}\left(\left(1 + \frac{\pi e}{4}\right) - 1\right)} = 1.$$

Hence, for every $t > 0$, we have

$$\mathbb{P}\left\{ \langle \theta, X_i \rangle^2 \leq t^2 \left\| \Sigma^{1/2}\theta \right\|^2 \right\} \leq \sqrt{\frac{8}{\pi e}\left(1 + \sqrt{1 + \frac{\pi e}{4}}\right)t} \leq 7^{1/4}t.$$

It confirms $X_i$ satisfies the small-ball condition with $C = 7^{1/4}$ and $\alpha = 1$. Application of Theorem 13 yields the desired claim. $\qquad\square$

*Proof of Lemma 5.* For a positive random variable $X$, we can express the expected value of $X$ as $\mathbf{E}[X] = \int_0^\infty \mathbb{P}\{X > t\}dt$. Applying this to our problem, we obtain

$$\mathbf{E}\left[ \lambda_{\max}\left( \left(\frac{1}{n}X^\top X\right)^{-1} \right) \right] = \int_0^\infty \mathbb{P}\left\{ \lambda_{\max}\left( \left(\frac{1}{n}X^\top X\right)^{-1} \right) > t \right\}dt$$

$$= \int_0^\infty \mathbb{P}\left\{ \lambda_{\min}\left( \left(\frac{1}{n}X^\top X\right) \right) < t^{-1} \right\}dt.$$

Now, let us set $C = \frac{21e^{10}}{\sigma_X^2}$. Using the previous result, we can rewrite the expectation of interest as

$$\mathbf{E}\left[ \lambda_{\max}\left( \left(\frac{1}{n}X^\top X\right)^{-1} \right) \right] = C + \int_C^\infty \mathbb{P}\left\{ \lambda_{\min}\left( \left(\frac{1}{n}X^\top X\right) \right) < t^{-1} \right\}dt$$

$$\leq C + \int_C^\infty \left(Ct^{-1}\right)^{n/6}dt$$

$$= C + C^{n/6}\left(1 - \frac{n}{6}\right)^{-1}\left(-C^{1-n/6}\right)$$

$$= C\left(1 + \frac{6}{n-6}\right),$$

which yields the claim. $\qquad\square$

**Proofs of theorems.** By utilizing the results of Lemmas 2 to 5, we provide the complete proofs for both Theorem 4 and Theorem 5.

*Proof of Theorem 4.* We begin by demonstrating that obtaining an upper bound on the expected error of the direction estimator can be reduced to finding a lower bound on the inner product $A_n = \langle \frac{\hat{\beta}}{\|\hat{\beta}\|}, \frac{\beta^*}{\|\beta^*\|} \rangle$. Specifically, a straightforward calculation yields

$$\mathbf{E}\left[ \left\| \frac{\hat{\beta}}{\|\hat{\beta}\|} - \frac{\beta^*}{\|\beta^*\|} \right\|^2 \right] = 2\left(1 - \mathbf{E}\left[ \left\langle \frac{\hat{\beta}}{\|\hat{\beta}\|}, \frac{\beta^*}{\|\beta^*\|} \right\rangle \right]\right). \tag{24}$$

Therefore, it suffices to establish a lower bound on $\mathbf{E}[A_n]$.

Taking advantage of Lemma 3, we derive a lower bound on $A_n$. Let $r = (\hat{\beta} - \beta^*)^\top (\frac{n}{\sigma_\xi^2} \Sigma_n)(\hat{\beta} - \beta^*)$. From Lemma 3, it follows that

$$A_n \geq \sqrt{1 - \frac{\sigma_\xi^2 r}{\|\beta^*\|^2 n} \lambda_{\max}(\Sigma_n^{-1})},$$

provided that $r \leq n\|\beta^*\|^2/\sigma_\xi^2 \lambda_{\min}(\Sigma_n^{-1})$. Since $1 - \sqrt{1-x} \leq x$ for $x \in [0, 1]$, it follows that

$$1 - A_n \leq \frac{\sigma_\xi^2 r}{\|\beta^*\|^2 n} \lambda_{\max}(\Sigma_n^{-1}),$$

as long as $r \leq n\|\beta^*\|^2/\sigma_\xi^2 \lambda_{\max}(\Sigma_n^{-1})$.

Next, we derive an upper bound on the expectation of $1 - A_n$. Noting that conditioned on $X$, $r$ follows the $\chi^2$ distribution with degree of freedom $d$, we have

$$\mathbf{E}[1 - A_n | X]$$

$$=\mathbf{E}\left[\left(\mathbb{1}\left\{r \leq \frac{n\|\beta^*\|^2}{\sigma_\xi^2 \lambda_{\max}(\Sigma_n^{-1})}\right\} + \mathbb{1}\left\{r > \frac{n\|\beta^*\|^2}{\sigma_\xi^2 \lambda_{\max}(\Sigma_n^{-1})}\right\}\right)(1 - A_n)\bigg| X\right]$$

$$\leq\mathbf{E}\left[\frac{\sigma_\xi^2 r}{\|\beta^*\|^2 n} \lambda_{\max}(\Sigma_n^{-1})\bigg| X\right] + \mathbb{P}\left\{r > \frac{n\|\beta^*\|^2}{\sigma_\xi^2 \lambda_{\max}(\Sigma_n^{-1})}\bigg| X\right\}$$

$$\leq 2\mathbf{E}\left[\frac{\sigma_\xi^2 r}{\|\beta^*\|^2 n} \lambda_{\max}(\Sigma_n^{-1})\bigg| X\right] \tag{25}$$

$$=\frac{2\sigma_\xi^2 d}{\|\beta^*\|^2 n} \lambda_{\max}(\Sigma_n^{-1}), \tag{26}$$

where we use the Markov inequality to obtain Eq (25).

By utilizing Eq (24), an upper bound on the expected error can be obtained by deriving an upper bound on the expectation of Eq (26). The random variable in Eq (26) is $\lambda_{\max}(\Sigma_n^{-1})$, which allows us to derive the upper bound on the expected error by obtaining an upper bound on the expectation of $\lambda_{\max}(\Sigma_n^{-1})$. To accomplish this, we apply Lemma 5. The upper bound from Lemma 5 can be substituted into Eq (26), resulting in the claimed upper bound. □

*Proof of Theorem 5.* We first utilize Lemma 2 to get an upper bound on the squared error of the norm estimator. Let $r = (\hat{\beta} - \beta^*)^\top (\frac{n}{\sigma_\xi^2} \Sigma_n)(\hat{\beta} - \beta^*)$. From Lemma 2, we have

$$\left\|\hat{\beta} - \beta^*\right\|^2 \leq \frac{\sigma_\xi^2 r}{n} \lambda_{\max}(\Sigma_n^{-1}).$$

Application of the triangle and reverse triangle inequality yields

$$\|\beta^*\| - \sqrt{\frac{\sigma_\xi^2 r}{n} \lambda_{\max}(\Sigma_n^{-1})} \leq \left\|\hat{\beta}\right\| \leq \|\beta^*\| + \sqrt{\frac{\sigma_\xi^2 r}{n} \lambda_{\max}(\Sigma_n^{-1})},$$

equivalently

$$\left(\left\|\hat{\beta}\right\| - \|\beta^*\|\right)^2 \leq \frac{\sigma_\xi^2 r}{n} \lambda_{\max}(\Sigma_n^{-1}).$$

Taking expectation conditioned on $X$ yields

$$\mathbf{E}\left[\left(\|\hat{\beta}\| - \|\beta^*\|\right)^2 \bigg| X\right]$$

$$=\mathbf{E}\left[\frac{\sigma_\xi^2 r}{n} \lambda_{\max}(\Sigma_n^{-1})\bigg| X\right]$$

$$\leq\frac{\sigma_\xi^2 d}{n} \lambda_{\max}(\Sigma_n^{-1}), \tag{27}$$

where we use the fact that $r$ follows the $\chi^2$ distribution with degree of freedom $d$ to obtain the last line. Again, application of Lemma 5 into expectation of Eq (27) yields the claim. □

# F  Details of Upper Bound Analyses

This section presents a detailed proof of the upper bound stated in Theorem 1, which is achieved through an analysis of the estimator constructed in Section 5. Specifically, we establish the following theorem:

**Theorem 14.** *Let $\hat{\beta}_n$ be the estimator constructed in Section 5. Then, there exists a universal constant $C > 0$ such that for any $\delta \in (0,1)$ and $n \geq 12(3d \vee 4\ln(M/\delta))/\min_{s \in [M]} p_s$,*

$$\mathcal{E}_n\left(\frac{1}{2}, \delta\right) \leq C \frac{\sigma_\xi^2 B^2 dM \vee \sigma_X^2 B^2 M \vee B^2 U^2}{n} + o\left(\frac{1}{n}\right).$$

To establish the validity of Theorem 14, we begin by proving Theorem 3, which demonstrates that the estimation error can be decomposed into the sum of errors associated with individual components. Subsequently, we derive upper bounds for the estimation errors of each component. Finally, we synthesize these results to provide a proof of Theorem 14.

## F.1  Proof of Theorem 3

We commence the error analysis of our estimator by decomposing the estimation error, as presented in Theorem 3. Recall the statement of Theorem 3

**Theorem 15.** *For the estimator defined in Eq (6), the mean square deviation from $f_{\mathrm{DP}}^*$ is bounded above by*

$$\sum_{s \in [M]} p_s \mathbf{E}\left[\left(\mathbf{E}\left[\widehat{\|\beta_{\cdot}\|}^2 \Big| n_{\cdot}\right]\right)^{1/2} \mathbf{E}\left[\left\langle \tilde{\beta}_s, \mu_s - \hat{\mu}_s \right\rangle^2 \Big| n_{\cdot}\right]^{1/2} + \sigma_X \mathbf{E}\left[\left(\widehat{\|\beta_{\cdot}\|} - \overline{\|\beta_{\cdot}^*\|}\right)^2 \Big| n_{\cdot}\right]^{1/2} + \right.$$

$$\sigma_X \overline{\|\beta_{\cdot}^*\|} \mathbf{E}\left[\left\|\tilde{\beta}_s - \beta_s^* / \|\beta_s^*\|\right\|^2 \Big| n_{\cdot}\right]^{1/2} + \mathbf{E}\left[\left(\sum_{s' \in [M]} \hat{p}_{s'} \left\langle \hat{\beta}_{s'}' - \beta_{s'}^*, \hat{\mu}_{s'}' \right\rangle\right)^2 \Big| n_{\cdot}\right]^{1/2} + $$

$$\left. \mathbf{E}\left[\left(\sum_{s' \in [M]} \hat{p}_{s'} \langle \beta_{s'}^*, \hat{\mu}_{s'}' - \mu_{s'} \rangle\right)^2 \Big| n_{\cdot}\right]^{1/2} + \left|\sum_{s' \in [M]} (\hat{p}_{s'} - p_{s'}) \langle \beta_{s'}^*, \mu_{s'} \rangle\right|\right)^2\right].$$

We provide a proof of Theorem 15 as follows:

*Proof of Theorem 15.* We begin by decomposing $\hat{f}_n(X, S) - f_{\mathrm{DP}}^*(X, S)$ into six terms. Recall the definitions of $\hat{f}_n(x, s)$ and $f_{\mathrm{DP}}^*(x, s)$:

$$\hat{f}_n(x, s) = \widehat{\|\beta_{\cdot}\|}\left\langle \tilde{\beta}_s, x - \hat{\mu}_s \right\rangle + \sum_{s' \in [M]} \hat{p}_{s'} \left\langle \hat{\beta}_s', \hat{\mu}_s' \right\rangle$$

$$f_{\mathrm{DP}}^*(x, s) = \overline{\|\beta_{\cdot}^*\|}\left\langle \frac{\beta_s^*}{\|\beta_s^*\|}, x - \mu_s \right\rangle + \sum p_{s'} \langle \beta_{s'}^*, \mu_{s'} \rangle.$$

Through elementary calculations, we obtain:

$$\hat{f}_n(X, S) - f_{\mathrm{DP}}^*(X, S) = \widehat{\|\beta_{\cdot}\|}\left\langle \tilde{\beta}_S, \mu_S - \hat{\mu}_S \right\rangle + \left(\widehat{\|\beta_{\cdot}\|} - \|\beta_{\cdot}^*\|\right)\left\langle \tilde{\beta}_S, X - \mu_S \right\rangle$$

$$+ \|\beta_{\cdot}^*\|\left\langle \tilde{\beta}_S - \frac{\beta_S^*}{\|\beta_S^*\|}, X - \mu_S \right\rangle + \sum_{s' \in [M]} \hat{p}_{s'} \left\langle \hat{\beta}_{s'}' - \beta_{s'}^*, \hat{\mu}_{s'}' \right\rangle$$

$$+ \sum_{s' \in [M]} \hat{p}_{s'} \langle \beta_{s'}^*, \hat{\mu}_{s'}' - \mu_{s'} \rangle + \sum_{s' \in [M]} (\hat{p}_{s'} - p_{s'}) \langle \beta_{s'}^*, \mu_{s'} \rangle. \quad (28)$$

By the Cauchy-Schwarz inequality, for two random variable $Z_1$ and $Z_2$, we have $\mathbf{E}[(Z_1 + Z_2)^2]^{1/2} \leq \mathbf{E}[Z_1^2]^{1/2} + \mathbf{E}[Z_2^2]^{1/2}$. By applying this fact into the expectation of Eq (28) conditioned on $S$ and $n.$ multiple times, we have

$$\mathbf{E}\left[\left(\hat{f}_n(X, S) - f_{\mathrm{DP}}^*(X, S)\right)^2\right]$$

$$= \sum_{s \in [M]} p_s \mathbf{E}\left[\mathbf{E}\left[\left(\hat{f}_n(X, S) - f_{\mathrm{DP}}^*(X, S)\right)^2 \Big| S = s, n.\right] \Big| S = s\right]$$

$$\leq \sum_{s \in [M]} p_s \mathbf{E}\left[\left(\mathbf{E}\left[\left(\widehat{\|\beta_\cdot\|}\left\langle \tilde{\beta}_S, \mu_S - \hat{\mu}_S\right\rangle\right)^2 \Big| S = s, n.\right]^{1/2}\right.\right. \tag{29}$$

$$+ \mathbf{E}\left[\left(\left(\widehat{\|\beta_\cdot\|} - \|\beta_\cdot^*\|\right)\left\langle \tilde{\beta}_S, X - \mu_S\right\rangle\right)^2 \Big| S = s, n.\right]^{1/2}$$

$$+ \mathbf{E}\left[\left(\|\beta_\cdot^*\|\left\langle \tilde{\beta}_S - \frac{\beta_S^*}{\|\beta_S^*\|}, X - \mu_S\right\rangle\right)^2 \Big| S = s, n.\right]^{1/2}$$

$$+ \mathbf{E}\left[\left(\sum_{s' \in [M]} \hat{p}_{s'}\left\langle \hat{\beta}_{s'}' - \beta_{s'}^*, \hat{\mu}_{s'}'\right\rangle\right)^2 \Big| S = s, n.\right]^{1/2}$$

$$+ \mathbf{E}\left[\left(\sum_{s' \in [M]} \hat{p}_{s'}\langle \beta_{s'}^*, \hat{\mu}_{s'}' - \mu_{s'}\rangle\right)^2 \Big| S = s, n.\right]^{1/2}$$

$$+ \left.\left.\mathbf{E}\left[\left(\sum_{s' \in [M]} (\hat{p}_{s'} - p_{s'})\langle \beta_{s'}^*, \mu_{s'}\rangle\right)^2 \Big| S = s, n.\right]^{1/2}\right)^2 \Big| S = s\right].$$

In the subsequent analyses, we derive upper bounds for each term in Eq (29).

(First term in Eq (29)) Due to the splitting of the sample, $\widehat{\|\beta_\cdot\|}$, $\tilde{\beta}_s$, and $\hat{\mu}_s$ are independent conditioned on $n.$. Thus, we have:

$$\mathbf{E}\left[\left(\widehat{\|\beta_\cdot\|}\left\langle \tilde{\beta}_S, \mu_S - \hat{\mu}_S\right\rangle\right)^2 \Big| S = s, n.\right]^{1/2}$$

$$= \mathbf{E}\left[\left(\widehat{\|\beta_\cdot\|}\left\langle \tilde{\beta}_s, \mu_s - \hat{\mu}_s\right\rangle\right)^2 \Big| n.\right]^{1/2}$$

$$= \mathbf{E}\left[\left(\widehat{\|\beta_\cdot\|}^2\right) \Big| n.\right]^{1/2} \mathbf{E}\left[\left\langle \tilde{\beta}_s, \mu_s - \hat{\mu}_s\right\rangle^2 \Big| n.\right]^{1/2}.$$

This term matches the first term of the desired bound.

(Second term in Eq (29)) Since $\widehat{\|\beta_\cdot\|}$, $\tilde{\beta}_s$, and $X$ are independent conditioned on $n.$, we have

$$\mathbf{E}\left[\left(\left(\widehat{\|\beta_\cdot\|} - \overline{\|\beta_\cdot^*\|}\right)\left\langle \tilde{\beta}_s, X - \mu_s\right\rangle\right)^2 \Big| S = s, n.\right]$$

$$= \sigma_X^2 \mathbf{E}\left[\left(\widehat{\|\beta_\cdot\|} - \overline{\|\beta_\cdot^*\|}\right)^2 \Big| n.\right],$$

where we use the fact that $X - \mu_s \sim N(0, \sigma_X^2 I)$ conditioned on $S = s$, and $\tilde{\beta}_s \in \mathcal{S}_{d-1}$ almost surely. This result corresponds to the second term of the desired bound.

(Third term in Eq (29)) Since $X - \mu_s \sim N(0, \sigma_X^2 I)$, we have

$$\mathbf{E}\left[\left(\overline{\|\beta_\cdot^*\|}\left\langle \tilde{\beta}_s - \frac{\beta_s^*}{\|\beta_s^*\|}, X - \mu_s\right\rangle\right)^2 \Big| S = s, n.\right]$$

$$=\sigma_X^2 \overline{\|\beta^*\|}^2 \mathbf{E}\left[\left\|\tilde{\beta}_s - \frac{\beta_s^*}{\|\beta_s^*\|}\right\|^2 \Big| n_\cdot\right],$$

which corresponds to the third term of the desired bound.

(Forth and fifth terms in Eq (29)) These terms are independent of $S$, so we can omit $S = s$ from the condition, resulting in the fourth and fifth terms of the desired bound.

(Sixth term in Eq (29)) This term does not contain any random variable when $n_\cdot$ is fixed. Thus, we can remove the expectation, yielding the sixth term of the desired bound. □

## F.2 Estimation Error Analyses for Each Component

This subsection presents an analysis of the estimation errors associated with each component estimator. In particular, we investigate the estimation errors of $\hat{\mu}_s$, $\widehat{\|\beta_\cdot\|}$, $\tilde{\beta}_s$, $\hat{\beta}_s'$, and $\hat{\mu}_s'$.

### F.2.1 Estimation Error Analysis for $\hat{\mu}_s$

Here, we presents the proof of the following theorem.

**Theorem 16.** *Given $s \in [M]$, if $n_s > 18d$, we have*

$$\sup_{v \in \mathbb{S}_{d-1}} \mathbf{E}\left[\langle v, \mu_s - \hat{\mu}_s\rangle^2 \Big| n_\cdot\right] \leq \frac{6\sigma_X^2}{n_s}.$$

*Proof of Theorem 16.* Given $v \in \mathbb{S}_{d-1}$, we have

$$\mathbf{E}\left[\langle v, \mu_s - \hat{\mu}_s\rangle^2 \Big| n_\cdot\right] = \left\langle v, \mathbf{E}\left[(\mu_s - \hat{\mu}_s)(\mu_s - \hat{\mu}_s)^\top \Big| n_\cdot\right] v\right\rangle.$$

According to the definition, $\hat{\mu}_s$ is an average of $n_{3,s}$ i.i.d. random variables following $N(\mu_s, \sigma_X^2 I)$. Hence, we have $\mu_s - \hat{\mu}_s \sim N(0, \frac{\sigma_X^2}{n_{3,s}} I)$, which implies $\mathbf{E}[(\mu_s - \hat{\mu}_s)(\mu_s - \hat{\mu}_s)^\top | n_\cdot] = \frac{\sigma_X^2}{n_{3,s}} I$. Consequently, we obtain:

$$\begin{aligned}\mathbf{E}\left[\langle v, \mu_s - \hat{\mu}_s\rangle^2 \Big| n_\cdot\right] &= \left\langle v, \frac{\sigma_X^2}{n_{3,s}} I v\right\rangle \\ &= \frac{\sigma_X^2}{n_{3,s}}\langle v, v\rangle = \frac{\sigma_X^2}{n_{3,s}}.\end{aligned}$$

Since $n_{3,s} \geq \lfloor n_s/3\rfloor \geq n_s/6$ for $n_s \geq 6$, the claim follows. □

### F.2.2 Estimation Error Analysis for $\widehat{\|\beta_\cdot\|}$

Here, we present the proof of the following theorem.

**Theorem 17.** *For any $s \in [M]$, we have*

$$\mathbf{E}\left[\left(\widehat{\|\beta_\cdot\|} - \overline{\|\beta_\cdot^*\|}\right)^2 \Big| n_\cdot\right]^{1/2} \leq \sqrt{\frac{189e^{10}\sigma_\xi^2 Md}{\sigma_X^2 n} + \sum_{s \in [M]} \mathbb{1}\{n_s \leq 18d\}\frac{n_s}{n}} + \left|\sum_{s \in [M]} (\hat{p}_s - p_s)\|\beta_s^*\|\right|.$$

*Proof of Theorem 17.* By combining the definitions of $\widehat{\|\beta_\cdot\|}$ and $\overline{\|\beta_\cdot^*\|}$ and utilizing the Cauchy-Schwarz inequality, we obtain

$$\mathbf{E}\left[\left(\widehat{\|\beta_\cdot\|} - \overline{\|\beta_\cdot^*\|}\right)^2 \Big| n_\cdot\right]^{1/2}$$

$$=\mathbf{E}\left[\left(\sum_{s \in [M]} \hat{p}_s\left(\widehat{\|\beta_s\|} - \overline{\|\beta_s^*\|}\right) + \sum_{s \in [M]} (\hat{p}_s - p_s)\overline{\|\beta_s^*\|}\right)^2 \Big| n_\cdot\right]^{1/2}$$

$$\leq \mathbf{E}\left[\left(\sum_{s\in[M]}\hat{p}_s\left(\|\widehat{\beta_s}\| - \overline{\|\beta_s^*\|}\right)\right)^2\bigg|n.\right]^{1/2} + \left|\sum_{s\in[M]}(\hat{p}_s - p_s)\overline{\|\beta_s^*\|}\right|. \tag{30}$$

Next, we derive an upper bound for the first term in Eq (30). Applying Jensen's inequality, we have:

$$\mathbf{E}\left[\left(\sum_{s\in[M]}\hat{p}_s\left(\|\widehat{\beta_s}\| - \overline{\|\beta_s^*\|}\right)\right)^2\bigg|n.\right]$$

$$\leq \sum_{s\in[M]}\frac{n_s}{n}\mathbf{E}\left[\left(\|\widehat{\beta_s}\| - \overline{\|\beta_s^*\|}\right)^2\bigg|n.\right]$$

$$= \sum_{s\in[M]}\frac{n_s}{n}\left(\mathbb{1}\{n_s > 18d\}\mathbf{E}\left[\left(\|\widehat{\beta_s}\| - \overline{\|\beta_s^*\|}\right)^2\bigg|n.\right] + \mathbb{1}\{n_s \leq 18d\}\|\beta_s^*\|^2\right)$$

Using the fact that $n_{1,s} > 6d$ for $n_s > 18d$ and employing Theorem 5, we obtain:

$$\mathbf{E}\left[\left(\sum_{s\in[M]}\hat{p}_s\left(\|\widehat{\beta_s}\| - \overline{\|\beta_s^*\|}\right)\right)^2\bigg|n.\right]$$

$$\leq \sum_{s\in[M]}\frac{n_s}{n}\frac{21e^{10}\sigma_\xi^2 d}{n_{1,s}}\left(1 + \frac{6}{n_{1,s} - 6}\right) + \sum_{s\in[M]}\mathbb{1}\{n_s \leq 18d\}\frac{\|\beta_s^*\|^2 n_s}{n}$$

$$\leq \frac{189e^{10}\sigma_\xi^2 Md}{\sigma_X^2 n} + \sum_{s\in[M]}\mathbb{1}\{n_s \leq 18d\}\frac{B^2 n_s}{n},$$

where the last line follows from the fact that $n_{1,s} \geq \lfloor n_s/3 \rfloor \geq n_s/6$ for $n_s \geq 6$, $\frac{6}{n_{1,s}-6} \leq 1/2$ for $n_s \geq 18$, and $\|\beta_s^*\| \leq B$. $\qquad\square$

### F.2.3 Estimation Error Analysis for $\tilde{\beta}_s$

Here, we will prove the following theorem.

**Theorem 18.** *For any $s \in [M]$, we have*

$$\mathbf{E}\left[\left\|\tilde{\beta}_s - \frac{\beta_s^*}{\|\beta_s^*\|}\right\|^2\bigg|n.\right] \leq \begin{cases} \dfrac{756e^{10}\sigma_\xi^2 d}{\sigma_X^2\|\beta_s^*\|^2 n_s} & \text{if } n_s > 18d, \\ 1 & \text{otherwise .} \end{cases}$$

*Proof of Theorem 18.* If $n_s \leq 18d$, $\tilde{\beta}_s = 0$, and we thus have $\mathbf{E}[\|\tilde{\beta}_s - \frac{\beta_s^*}{\|\beta_s^*\|}\|^2|n.] = \|\frac{\beta_s^*}{\|\beta_s^*\|}\|^2 = 1$. For $n_s > 18d$, we have $n_{2,s} > 6d$. Application of Theorem 4 yields

$$\mathbf{E}\left[\left\|\tilde{\beta}_s - \frac{\beta_s^*}{\|\beta_s^*\|}\right\|^2\bigg|n.\right] \leq \frac{84e^{10}\sigma_\xi^2 d}{\sigma_X^2\|\beta_s^*\|^2 n_{2,s}}\left(1 + \frac{6}{n_{2,s} - 6}\right)$$

We get the claim in the same manner as the proof of Theorem 17. $\qquad\square$

### F.2.4 Estimation Error Analysis for $\hat{\beta}'_{s'}$

Here, we will prove the following theorem.

**Theorem 19.** *Given $s \in [M]$, let $\Sigma_s = \mathbf{E}[X_s X_s^\top]$ for $X_s \sim N(\mu_s, \sigma_X^2 I)$. Then, if $n_s > 12d$, we have*

$$\mathbf{E}\left[\left\|\Sigma_s^{1/2}\left(\hat{\beta}'_s - \beta_s^*\right)\right\|^2\bigg|n.\right] \leq \frac{4\sigma_\xi^2 d}{n_s} + 504e^{10}\sigma_\xi^2\left(\frac{4d}{n_s}\right)^2.$$

To prove Theorem 19, we utilize the following theorem presented by Mourtada [15].

**Theorem 20** (Theorem 3 in [15]). *Let $X$ be a random vector in $\mathbb{R}^d$ such that it statisfies the small-ball condition of Eq (19) and $\mathbf{E}[\|\Sigma^{-1/2}X\|^4] \leq \kappa d$ for some $\kappa > 0$, where $\Sigma = \mathbf{E}[XX^\top]$. Let $\hat{\Sigma}_n = \frac{1}{n}\sum_{i=1}^n X_i X_i^\top$, where $X_i$ are i.i.d. copies of $X$. If $n \geq 6\alpha^{-1}d \wedge 12\alpha^{-1}\ln(12\alpha^{-1})$,*

$$\frac{1}{n}\mathbf{E}\Big[\mathrm{Tr}\Big(\hat{\Sigma}_n^{-1}\Sigma\Big)\Big] \leq \frac{d}{n} + 8C'\kappa\left(\frac{d}{n}\right)^2,$$

*where $\alpha$ and $C'$ are as in Theorem 13.*

*Proof of Theorem 19.* We can easily confirm that $\hat{\beta}'_s \sim N(\beta_s^*, \frac{\sigma_\xi^2}{n'_{1,s}}(\Sigma'_{1,s})^{-1})$ conditioned on $n.$ and $X'_{1,s}$, where $\Sigma'_{1,s} = \frac{1}{n'_{1,s}}X'_{1,s}(X'_{1,s})^\top$. Noting that $\mathbf{E}[(\hat{\beta}'_s - \beta_s^*)(\hat{\beta}'_s - \beta_s^*)^\top|X, n.] = \frac{\sigma_\xi^2}{n'_{1,s}}(\Sigma'_{1,s})^{-1}$, we have

$$\mathbf{E}\Big[\big\|\Sigma_{m,s}^{1/2}\big(\hat{\beta}'_{s'} - \beta_s^*\big)\big\|^2\Big|n.\Big]$$
$$=\mathrm{Tr}\Big(\Sigma_s \mathbf{E}\Big[\big(\hat{\beta}'_{s'} - \beta_s^*\big)\big(\hat{\beta}'_{s'} - \beta_s^*\big)^\top\Big|n.\Big]\Big)$$
$$=\frac{\sigma_\xi^2}{n'_{1,s}}\mathbf{E}\Big[\mathrm{Tr}\Big(\Sigma_s\big(\Sigma'_{1,s}\big)^{-1}\Big)\Big|n.\Big]. \tag{31}$$

We apply Theorem 20 to the expected trace term in Eq (31). To do so, we need to check $X'_{1,s}$ satisfies the small-ball condition of Eq (19) and the kurtosis condition $\mathbf{E}[\|\Sigma_s^{-1/2}X'_{1,s}\|^4] \leq \kappa d$.

The small-ball condition is confirmed by the same manner in the proof of Lemma 5, with $C = 7^{1/4}$ and $\alpha = 1$. Here, we prove the satisfaction of the kurtosis condition. For a multivariate Gaussian random variable $X \sim N(\mu, \Lambda)$ such that $\lambda_{\min}(\Lambda) > 0$, $\Sigma := \mathbf{E}[XX^\top] = \mu\mu^\top + \Lambda$, and $\|\Sigma^{-1/2}X\|^4 = \langle X, \Sigma^{-1}X\rangle$

$$\mathbf{E}\Big[\big\|\Sigma^{-1/2}X\big\|^4\Big] = \mathbf{E}\big[\langle X, \Sigma^{-1}X\rangle^2\big]$$
$$=\mathbf{Var}\big[\langle X, \Sigma^{-1}X\rangle\big] + \big(\mathbf{E}\big[\langle X, \Sigma^{-1}X\rangle\big]\big)^2.$$

Since we have

$$\mathbf{E}\big[\langle X, \Sigma^{-1}X\rangle\big] = \mathrm{Tr}\big(\Sigma^{-1}\big(\mu\mu^\top + \Lambda\big)\big) = d$$
$$\mathbf{Var}\big[\langle X, \Sigma^{-1}X\rangle\big] = 2\mathrm{Tr}\big(\Sigma^{-1}\Lambda\Sigma^{-1}\big(\mu\mu^\top + \Lambda\big)\big) + 2\mathrm{Tr}\big(\Sigma^{-1}\Lambda\Sigma^{-1}\mu\mu^\top\big)$$
$$=2\mathrm{Tr}\big(\Sigma^{-1}\Lambda\big) + 2\mathrm{Tr}\big(\Sigma^{-1}\Lambda\Sigma^{-1}\mu\mu^\top\big)$$
$$=2\mathrm{Tr}\big(\Sigma^{-1}\big(\mu\mu^\top + \Lambda\big)\big) - 2\mathrm{Tr}\big(\mu\mu^\top\Sigma^{-1}\mu\mu^\top\big)$$
$$=2d - 2\|\mu\|^2\langle\mu, \Sigma^{-1}\mu\rangle \leq 2d.$$

Hence, the kurtosis condition satisfies with $\kappa = 3$.

Application of Theorem 20 into Eq (31) yields

$$\mathbf{E}\Big[\big\|\Sigma_{m,s}^{1/2}\big(\hat{\beta}'_{s'} - \beta_s^*\big)\big\|^2\Big|n.\Big] \leq \frac{\sigma_\xi^2 d}{n'_{1,s}} + 504e^{10}\sigma_\xi^2\left(\frac{d}{n'_{1,s}}\right)^2,$$

provided that $n_s \geq 12d$. We get the claim from the fact that $n'_{1,s} \geq \lfloor n_s/2 \rfloor \geq n_s/4$ for $n_s \geq 4$,. $\square$

### F.2.5 Estimation Error Analysis for $\hat{\mu}'_s$

Here, we will prove the following theorem.

**Theorem 21.** *Given $s \in [M]$ and $v \in \mathbb{R}^d$, if $n_s > 12d$, we have*

$$\mathbf{E}\Big[\langle v, \hat{\mu}'_s - \mu_s\rangle^2\Big|n.\Big] \leq \frac{4\sigma_X^2\|v\|^2}{n_s}.$$

*Proof of Theorem 21.* By definition, we have $\hat{\mu}'_s \sim N(\mu_s, \frac{\sigma_X^2}{n'_{2,s}}I)$ conditioned on $n.$ for $n_s > 12d$. Hence, we have

$$\mathbf{E}\left[\langle v, \hat{\mu}'_s - \mu_s\rangle^2 \Big| n.\right] \leq \frac{\sigma_X^2 \|v\|^2}{n'_{2,s}}.$$

We get the claim following the same manner of the proof of Theorem 19. $\qquad\square$

### F.3 Some Auxiliary Lemmas

This subsections introduce some auxiliary lemmas for use to prove Theorem 14. Specifically, we demonstrate the following lemmas:

**Lemma 6.** *Let $a_1, ..., a_M \in \mathbb{R}$ be arbitrary numbers. Then, we have*

$$\mathbf{E}\left[\left(\sum_{s \in [M]} a_s(\hat{p}_s - p_s)^2\right)\right] = \frac{1}{n}\mathbf{Var}[a_S].$$

**Lemma 7.** *For a constant $c > 0$, we have for any $s \in [M]$*

$$\mathbf{E}\left[n_s^{-1} \mathbb{1}\{n_s > c\}\right] \leq \frac{1 + c^{-1}}{p_s(n+1)}.$$

**Lemma 8.** *Let $c > 0$ be a constant. If $n > 2c/\min_{s \in [M]} p_s$, we have for any $s \in [M]$*

$$\mathbb{P}\{n_s \leq c\} \leq e^{-np_s/8}.$$

*Proof of Lemma 6.* Since $n\hat{p}.$ follows the multinomial distribution with the parameters $n$ and $p.$, using the variance and covariance of the multinomial distribution, we have

$$\mathbf{E}\left[\left(\sum_{s \in [M]} a_s(\hat{p}_s - p_s)\right)^2\right]$$

$$= \sum_{s \in [M]} \frac{a_s^2 p_s(1 - p_s)}{n} - \sum_{s,s' \in [M]:s \neq s'} \frac{a_s a_{s'} p_s p_{s'}}{n}$$

$$= \sum_{s \in [M]} \frac{a_s p_s}{n}\left(a_s - \sum_{s' \in [M]} a_{s'} p_{s'}\right).$$

Let $\bar{a} = \sum_{s \in [M]} p_s a_s$. Then, we have

$$\sum_{s \in [M]} a_s p_s(a_s - \bar{a})$$

$$= \sum_{s \in [M]} p_s(a_s - \bar{a})^2 + \sum_{s \in [M]} \bar{a} p_s(a_s - \bar{a})$$

$$= \sum_{s \in [M]} p_s(a_s - \bar{a})^2 = \mathbf{Var}[a_S].$$

Hence,

$$\mathbf{E}\left[\left(\sum_{s \in [M]} a_s(\hat{p}_s - p_s)\right)^2\right] = \frac{1}{n}\mathbf{Var}[a_S].$$

$\qquad\square$

*Proof of Lemma 7.* For a random variable $X$ following the binomial distribution with the parameters $n$ and $p$, $\mathbf{E}[\frac{1}{X+1}] = \frac{1}{p(n+1)}(1-(1-p)^{n+1})$ [5]. Since $n_s$ follows the binomial distribution with the parameters $n$ and $p_s$, we have We have

$$
\mathbf{E}\big[n_s^{-1}\,\mathbb{1}\{n_s > c\}\big]
$$
$$
=\mathbf{E}\bigg[\frac{1}{n_s + 1}\bigg(1 + \frac{1}{n_s}\bigg)\mathbb{1}\{n_s > c\}\bigg]
$$
$$
\leq (1 + c^{-1})\mathbf{E}\bigg[\frac{1}{n_s + 1}\bigg]
$$
$$
=\frac{1 + c^{-1}}{p_s(n+1)}\big(1 - (1 - p_s)^{n+1}\big) \leq \frac{1 + c^{-1}}{p_s(n+1)}.
$$

$\square$

*Proof of Lemma 8.* From the Chernoff bound, we have

$$
\mathbb{P}\{n_s \leq c\} \leq \exp\bigg(-\frac{n}{2p_s}\Big(\frac{c}{n} - p_s\Big)^2\bigg).
$$

Under the assumption, we have $c \leq np_s/2$. Then, we have $\frac{n}{2p_s}\big(\frac{c}{n} - p_s\big)^2 \geq n/8$, which gives the claim. $\square$

## F.4  Proof of Theorem 14

*Proof of Theorem 14.* We begin by characterizing the estimation error by each component's estimation error shown in Theorems 16 to 19 and 21. Specifically, we characterize the estimation error using the following error terms:

$$
e_{\mathrm{mean},s}^2 = \sup_{v\in\mathbb{S}_{d-1}} \mathbf{E}\Big[\langle v, \mu_s - \hat{\mu}_s\rangle^2\big|n.\Big]
$$
$$
e_{\mathrm{norm}}^2 = \mathbf{E}\big[(\overline{\|\beta_\cdot^*\|} - \widehat{\|\beta_\cdot\|})^2\big|n.\big]
$$
$$
e_{\mathrm{coef},s}^2 = \mathbf{E}\big[\|\tilde{\beta}_s - \frac{\beta_s^*}{\|\beta_s^*\|}\|^2\big|n.\big]
$$
$$
e_{\mathrm{coef}',s}^2 = \mathbf{E}\bigg[\Big\|\Sigma_s^{1/2}\big(\hat{\beta}_s' - \beta_s^*\big)\Big\|^2\big|n.\bigg]
$$
$$
e_{\mathrm{mean}',s}^2 = \sup_{v\in\mathbb{S}_{d-1}} \mathbf{E}\big[\langle \hat{\mu}_s' - \mu_s, v\rangle^2\big|n.\big]
$$
$$
e_{\mathrm{prob}}^2 = \bigg|\sum_{s'\in[M]} (\hat{p}_{s'} - p_{s'})\langle\beta_{s'}^*, \mu_{s'}\rangle\bigg|^2,
$$

where $\Sigma_x = \mathbf{E}[X_s X_s^\top]$ for $X_s \sim N(\mu_s, \sigma_X^2 I)$.

We analyze each term in Eq (7) one by one.

(First term in Theorem 3) Recall the first term in Theorem 3

$$
\mathbf{E}\Big[\widehat{\|\beta_\cdot\|}^2\big|n.\Big]^{1/2}\mathbf{E}\Big[\big\langle \tilde{\beta}_s, \mu_s - \hat{\mu}_s\big\rangle^2\big|n.\Big]^{1/2}.
$$

From the Cauchy–Schwarz inequality, we have

$$
\mathbf{E}\Big[\widehat{\|\beta_\cdot\|}^2\big|n.\Big]^{1/2} \leq \mathbf{E}\Big[\overline{\|\beta_\cdot^*\|}^2\big|n.\Big]^{1/2} + \mathbf{E}\bigg[\Big(\overline{\|\beta_\cdot^*\|} - \widehat{\|\beta_\cdot\|}\Big)^2\big|n.\bigg]^{1/2}
$$
$$
\leq B + e_{\mathrm{norm}}.
$$

By definition, $\tilde{\beta}_s = 0$ for $n_s \leq 18d$, which indicates that

$$\mathbf{E}\left[\left\langle\tilde{\beta}_s, \mu_s - \hat{\mu}_s\right\rangle^2 \Big| n.\right]^{1/2} = 0.$$

In the case of $n_s > 18d$, by utilizing the fact $\tilde{\beta}_s$ and $\hat{\mu}_s$ are independent conditioned on $n.$ due to the sample spilitting and $\tilde{\beta}_s \in \mathbb{S}_{d-1}$, we obtain

$$\mathbf{E}\left[\left\langle\tilde{\beta}_s, \mu_s - \hat{\mu}_s\right\rangle^2 \Big| n.\right] \leq \sup_{v \in \mathbb{S}_{d-1}} \mathbf{E}\left[\langle v, \mu_s - \hat{\mu}_s\rangle^2 \Big| n.\right] = e_{\mathrm{mean},s}^2.$$

Consequently, we have

$$\mathbf{E}\left[\left\|\widehat{\beta.}\right\|^2 \Big| n.\right]^{1/2} \mathbf{E}\left[\left\langle\tilde{\beta}_s, \mu_s - \hat{\mu}_s\right\rangle^2 \Big| n.\right]^{1/2} \leq (B + e_{\mathrm{norm}})e_{\mathrm{mean},s} \mathbb{1}\{n_s > 18d\}. \tag{32}$$

(Second term in Theorem 3) Recall the second term in Theorem 3

$$\sigma_X \mathbf{E}\left[\left(\left\|\widehat{\beta.}\right\| - \overline{\|\beta^*\|}\right)^2 \Big| n.\right]^{1/2}.$$

Using the notation of $e_{\mathrm{norm}}$, we have

$$\sigma_X \mathbf{E}\left[\left(\left\|\widehat{\beta.}\right\| - \overline{\|\beta^*\|}\right)^2 \Big| n.\right]^{1/2} = \sigma_X e_{\mathrm{norm}}. \tag{33}$$

(Third term in Theorem 3) Recall the third term in Theorem 3

$$\sigma_X \overline{\|\beta^*\|} \mathbf{E}\left[\left\|\tilde{\beta}_s - \frac{\beta_s^*}{\|\beta_s^*\|}\right\|^2 \Big| n.\right]^{1/2}.$$

Substituting $e_{\mathrm{coef},s}$ yields

$$\sigma_X \overline{\|\beta^*\|} \mathbf{E}\left[\left\|\tilde{\beta}_s - \frac{\beta_s^*}{\|\beta_s^*\|}\right\|^2 \Big| n.\right]^{1/2} = \sigma_X \overline{\|\beta^*\|} e_{\mathrm{coef},s}. \tag{34}$$

(Fourth term in Theorem 3) Recall the fourth term in Theorem 3

$$\mathbf{E}\left[\left(\sum_{s \in [M]} \hat{p}_s\left\langle\hat{\beta}_s' - \beta_s^*, \hat{\mu}_s'\right\rangle\right)^2 \Big| n.\right]^{1/2}.$$

Due to the sample splitting, $\hat{\beta}_s'$ and $\hat{\mu}_s'$ are mutually independent. Also, we have $\mathbf{E}[\langle\hat{\beta}_s' - \beta_s^*, \hat{\mu}_s'\rangle] = 0$. Hence,

$$\mathbf{E}\left[\left(\sum_{s \in [M]} \hat{p}_s\left\langle\hat{\beta}_s' - \beta_s^*, \hat{\mu}_s'\right\rangle\right)^2 \Big| n.\right]$$

$$= \sum_{s \in [M]} \hat{p}_s^2 \mathbf{E}\left[\left\langle\hat{\beta}_s' - \beta_s^*, \hat{\mu}_s'\right\rangle^2 \Big| n.\right]$$

$$= \sum_{s \in [M]} \hat{p}_s^2 \mathbf{E}\left[\left\langle\hat{\beta}_s' - \beta_s^*, \hat{\mu}_s'\right\rangle^2 \Big| n.\right] \mathbb{1}\{n_s > 12d\}.$$

Since $\mathbf{E}[(\hat{\mu}_s')(\hat{\mu}_s')^\top] = \mu\mu^\top + \frac{\sigma_X^2}{n_{2,s}'}I = \Sigma_s - (1 - \frac{1}{n_{2,s}'})\sigma_X^2 I$, we have

$$\mathbf{E}\left[\left(\sum_{s \in [M]} \hat{p}_s\left\langle\hat{\beta}_s' - \beta_s^*, \hat{\mu}_s'\right\rangle\right)^2 \Big| n.\right]$$

$$= \sum_{s\in[M]} \hat{p}_s^2 \mathbf{E}\left[\left\|\left(\Sigma_s - \left(1 - \frac{1}{n'_{2,s}}\right)\sigma_X^2 I\right)^{1/2}\left(\hat{\beta}'_s - \beta_s^*\right)\right\|^2 \Big| n.\right] \mathbb{1}\{n_s > 12d\}$$

$$\leq \sum_{s\in[M]} \hat{p}_s^2 \mathbf{E}\left[\left\|\Sigma_s^{1/2}\left(\hat{\beta}'_s - \beta_s^*\right)\right\|^2 \Big| n.\right] \mathbb{1}\{n_s > 12d\} = \sum_{s\in[M]} \hat{p}_s^2 e_{\text{coef}',s}^2 \mathbb{1}\{n_s > 12d\}, \qquad (35)$$

where we use the fact $\Sigma_s \succeq \Sigma_s - (1 - \frac{1}{n'_{2,s}})\sigma_X^2 I$, and for symmetric matrices $A$ and $B$ such that $A \succeq B$, $\langle v, Av\rangle \geq \langle v, Bv\rangle$ for any $v \in \mathbb{R}$. For $n_s \leq 18d$, the error is zero because $\hat{\mu}'_s = 0$

(Fifth term in Theorem 3) Recall the fifth term in Theorem 3

$$\mathbf{E}\left[\left(\sum_{s\in[M]} \hat{p}_s \langle \beta_s^*, \hat{\mu}'_s - \mu_s\rangle\right)^2 \Big| n.\right]^{1/2}.$$

Since $\hat{\mu}'_s$ are independent, we have

$$\mathbf{E}\left[\left(\sum_{s\in[M]} \hat{p}_s \langle \beta_s^*, \hat{\mu}'_s - \mu_s\rangle\right)^2 \Big| n.\right]$$

$$= \sum_{s\in[M]} \hat{p}_s^2 \mathbf{E}\left[\langle \beta_s^*, \hat{\mu}'_s - \mu_s\rangle^2 \Big| n.\right] \mathbb{1}\{n_s > 12d\}$$

$$+ \sum_{s,s'\in[M]} \hat{p}_s \hat{p}_{s'} \mathbb{1}\{n_s < 12d, n_{s'} < 12d\}\langle \beta_s^*, \mu_s\rangle\langle \beta_{s'}^*, \mu_{s'}\rangle$$

$$\leq \sum_{s\in[M]} \hat{p}_s^2 \|\beta_s^*\|^2 \sup_{v\in\mathbb{S}_{d-1}} \mathbf{E}\left[\langle v, \hat{\mu}'_s - \mu_s\rangle^2 \Big| n.\right] \mathbb{1}\{n_s > 12d\} + \frac{144d^2 M^2}{n^2}$$

$$\leq \sum_{s\in[M]} \hat{p}_s^2 B^2 e_{\text{mean}',s}^2 \mathbb{1}\{n_s > 12d\} + o\left(\frac{1}{n}\right). \qquad (36)$$

(Sixth term in Theorem 3) Recall the sixth term in Theorem 3

$$\left|\sum_{s'\in[M]} (\hat{p}_{s'} - p_{s'})\langle \beta_{s'}^*, \mu_{s'}\rangle\right|,$$

which is equivalent to $e_{\text{prob}}$.

By combining Theorem 3 and Eqs (32) to (36), we get

$$\mathbf{E}\left[\left(\hat{f}_n(X,S) - f_{\text{DP}}^*(X,S)\right)^2\right]$$

$$\leq \sum_{s\in[M]} p_s \mathbf{E}\left[\left((B + e_{\text{norm}})e_{\text{mean},s}\mathbb{1}\{n_s > 18d\} + \sigma_X e_{\text{norm}} + \sigma_X \overline{\|\beta_*^*\|} e_{\text{coef},s}\right.\right.$$

$$+ \left(\sum_{s'\in[M]} \hat{p}_{s'}^2 e_{\text{coef}',s'}^2 \mathbb{1}\{n_{s'} > 12d\}\right)^{1/2}$$

$$\left.\left.+ \left(\sum_{s'\in[M]} \hat{p}_{s'}^2 B^2 e_{\text{mean}',s'}^2 \mathbb{1}\{n_{s'} > 12d\} + o\left(\frac{1}{n}\right)\right)^{1/2} + e_{\text{prob}}\right)^2\right].$$

The triangle inequality gives that

$$\mathbf{E}\left[\left(\hat{f}_n(X,S) - f_{\mathrm{DP}}^*(X,S)\right)^2\right]$$

$$\leq \sum_{s\in[M]} 7p_s\Bigg(B^2\mathbf{E}\big[e_{\mathrm{mean},s}^2\,\mathbb{1}\{n_s > 18d\}\big] + \mathbf{E}\big[e_{\mathrm{norm}}^2\big]\mathbf{E}\big[e_{\mathrm{mean},s}^2\,\mathbb{1}\{n_s > 18d\}\big]$$

$$+ \sigma_X^2\mathbf{E}\big[e_{\mathrm{norm}}^2\big] + \sigma_X^2\overline{\|\beta_\cdot^*\|}^2\mathbf{E}\big[e_{\mathrm{coef},s}^2\big]\Bigg) + \sum_{s\in[M]} 7\Bigg(\mathbf{E}\big[\hat{p}_s^2 e_{\mathrm{coef}',s}^2\,\mathbb{1}\{n_s > 12d\}\big]$$

$$+ \mathbf{E}\big[\hat{p}_s^2 B^2 e_{\mathrm{mean}',s}^2\,\mathbb{1}\{n_s > 12d\}\big]\Bigg) + 7\mathbf{E}\big[e_{\mathrm{prob}}^2\big] + o\left(\frac{1}{n}\right).$$

By applying Theorem 16 and Lemma 7, we have

$$\mathbf{E}\big[e_{\mathrm{mean},s}^2\,\mathbb{1}\{n_s > 18d\}\big] \leq \mathbf{E}\left[\frac{6\sigma_X^2}{n_s}\,\mathbb{1}\{n_s > 18d\}\right] \leq \frac{6\sigma_X^2}{p_s(n+1)}\left(1 + \frac{1}{18d}\right).$$

Also, from Theorem 17 and Lemmas 6 and 8, we have

$$\mathbf{E}\big[e_{\mathrm{norm}}^2\big] \leq \mathbf{E}\left[\left(\sqrt{\frac{189e^{10}\sigma_\xi^2 Md}{\sigma_X^2 n} + \sum_{s\in[M]}\mathbb{1}\{n_s \leq 18d\}\frac{n_s}{n}} + \left|\sum_{s\in[M]}(\hat{p}_s - p_s)\|\beta_s^*\|\right|\right)^2\right]$$

$$\leq \frac{378e^{10}\sigma_\xi^2 Md}{\sigma_X^2 n} + \sum_{s\in[M]}\mathbf{E}\left[\mathbb{1}\{n_s \leq 18d\}\frac{2n_s}{n}\right] + \frac{2}{n}\mathbf{Var}(\|\beta_S^*\|)$$

$$\leq \frac{378e^{10}\sigma_\xi^2 Md}{\sigma_X^2 n} + \sum_{s\in[M]}\mathbb{P}\{n_s \leq 18d\}\frac{36d}{n} + \frac{2\max_s\|\beta_s^*\|^2}{n}$$

$$\leq \frac{378e^{10}\sigma_\xi^2 Md}{\sigma_X^2 n} + \sum_{s\in[M]}\frac{36d}{n}e^{-np_s/8} + \frac{2B^2}{n}$$

$$= \frac{378e^{10}\sigma_\xi^2 Md}{\sigma_X^2 n} + \frac{2B^2}{n} + o\left(\frac{1}{n}\right),$$

provided that $n > 36d/\min_{s\in[M]} p_s$. By utilizing Theorem 18 and Lemmas 7 and 8, we have

$$\mathbf{E}\big[e_{\mathrm{coef},s}^2\big] \leq \mathbf{E}\left[\frac{756e^{10}\sigma_\xi^2 d}{\sigma_X^2\|\beta_s^*\|^2 n_s}\,\mathbb{1}\{n_s > 18d\}\right] + \mathbb{P}\{n_s \leq 18d\}$$

$$\leq \frac{756e^{10}\sigma_\xi^2 d}{p_s\sigma_X^2\|\beta_s^*\|^2 n}\left(1 + \frac{1}{18d}\right) + e^{-np_s/8}$$

$$= \frac{756e^{10}\sigma_\xi^2 d}{p_s\sigma_X^2\|\beta_s^*\|^2 n}\left(1 + \frac{1}{18d}\right) + o\left(\frac{1}{n}\right),$$

provided that $n > 36d/\min_{s\in[M]} p_s$. Application of Theorem 19 gives

$$\mathbf{E}\big[\hat{p}_s^2 e_{\mathrm{coef}',s}^2\,\mathbb{1}\{n_s > 12d\}\big] \leq \mathbf{E}\left[\left(\frac{n_s}{n}\right)^2\left(\frac{4\sigma_\xi^2}{n_s} + 540e^{10}\sigma_\xi^2\left(\frac{4d}{n_s}\right)^2\right)\mathbb{1}\{n_s > 12d\}\right]$$

$$\leq \mathbf{E}\left[\frac{n_s}{n}\,\mathbb{1}\{n_s > 12d\}\right]\frac{4\sigma_\xi^2}{n_s} + o\left(\frac{1}{n}\right) \leq p_s\frac{4\sigma_\xi^2}{n_s} + o\left(\frac{1}{n}\right).$$

By Theorem 21, we have

$$\mathbf{E}\big[\hat{p}_s^2 e_{\mathrm{mean}',s}^2\,\mathbb{1}\{n_s > 12d\}\big] \leq \mathbf{E}\left[\frac{n_s}{n}\frac{4\sigma_X^2}{n}\,\mathbb{1}\{n_s > 12d\}\right] \leq p_s\frac{4\sigma_X^2}{n}.$$

Application of Lemma 6 with $a_s = \langle\beta_s^*, \mu_s\rangle$ into $e_{\mathrm{prob}}^2$ yields

$$\mathbf{E}\big[e_{\mathrm{prob}}^2\big] \leq \frac{1}{n}\mathbf{Var}(\langle\beta_S^*, \mu_S\rangle) \leq \frac{B^2 U^2}{n}.$$

Synthesizing the results so far, there exists an universal constant $C > 0$ such that

$$\mathbf{E}\left[\left(\hat{f}_n(X, S) - f^*_{\mathrm{DP}}(X, S)\right)^2\right]$$

$$\leq \sum_{s \in [M]} C p_s \left(\frac{\sigma_X^2 B^2}{p_s n} + \frac{\sigma_\xi^2 M d}{n} + \frac{\sigma_X^2 B^2}{n} + \frac{\sigma_\xi^2 \overline{\|\beta_\cdot^*\|}^2 d}{\|\beta_s^*\|^2 p_s n}\right)$$

$$+ \sum_{s \in [M]} C\left(\frac{p_s \sigma_X^2}{n} + \frac{p_s \sigma_X^2}{n}\right) + C\frac{B^2 U^2}{n} + o\left(\frac{1}{n}\right).$$

Consequently, there exists an universal constant $C > 0$ such that

$$\mathbf{E}\left[\left(\hat{f}_n(X, S) - f^*_{\mathrm{DP}}(X, S)\right)^2\right] \leq$$

$$C\left(\frac{\sigma_X^2 B^2 M}{n} + \frac{\sigma_\xi^2 M d}{n} + \frac{\sigma_X^2 B^2}{n} + \frac{\sigma_\xi^2 B^2 M d}{n} + \frac{\sigma_X^2}{n} + \frac{\sigma_X^2}{n} + \frac{B^2 U^2}{n}\right) + o\left(\frac{1}{n}\right).$$

Then, the dominating terms match the claim. □

## G  Details of Lower Bound Analyses

This section provides the proofs of the lower bound analyses results.

### G.1  Proof of Lower Bound in Theorem 1

**Theorem 22.** *If $M(d - 1) > 16$, there exists an universal constant $C > 0$ such that for any $\alpha > 0$ and $\delta \in (0, 1)$,*

$$\mathcal{E}_n(\alpha, \delta) \geq C\frac{\sigma_\xi^2 B^2 d M}{n} - o\left(\frac{1}{n}\right).$$

*Proof of Theorem 22.* The Varshamov-Gilbert bound guarantees that there exists a subset $\mathcal{V}' \subseteq \mathcal{V}$ such that $|\mathcal{V}'| \geq 2^{M(d-1)/8}$ and $d_H(v_s, v'_s) \geq (d - 1)/8$ for any $v, v' \in \mathcal{V}'$. With the choice of $\epsilon_s^2 = (\frac{d-1}{16} - \frac{1}{M})\sigma_\xi^2/2\sigma_X^2 B_s^2 n_s$, we confirm by Theorem 8 that $\inf_\pi \frac{1}{K} \sum_{v \in \mathcal{V}'} D_{\mathrm{KL}}\left(\pi_{\theta_v|n.}, \pi\right) \leq \max_{v,v' \in \mathcal{V}'} D_{\mathrm{KL}}\left(\pi_{\theta_{v'}|n.}, \pi\right) \leq \ln(|\mathcal{V}'|/4)/2 \geq M(d - 1)/16 - 1$. From Theorem 8 and the fact $d_H(v_s, v'_s) \geq (d - 1)/8$, we can apply Theorem 6 with $\epsilon = \sum_{s \in [M]} p_s \frac{(\sum_{s' \in [M]} p_s B_s)^2}{B_s^2}(\frac{d-1}{16} - \frac{1}{M})\sigma_\xi^2/2n_s$. From the fact that $\mathbf{E}[\frac{1}{n_s+1}] = \frac{1}{p_s(n+1)}(1 - (1 - p_s)^n)$ due to [5], there exists an universal constant $C > 0$ such that $\mathbf{E}[\frac{\epsilon}{2}] \geq C(\frac{1}{M}\sum_{s \in [M]} \frac{(\sum_{s' \in [M]} p_{s'} B_{s'})^2}{B_s^2})\frac{\sigma_\xi^2 M d}{n} - o(\frac{1}{n})$ We can get the claim by confirming that there exists $B_1, ..., B_M$ such that $(\frac{1}{M}\sum_{s \in [M]} \frac{(\sum_{s' \in [M]} p_{s'} B_{s'})^2}{B_s^2}) = B^2$ and $B_s \leq B$. Because for $B_2 = ... = B_M = B$, tending $B_1$ to 0 results in $(\frac{1}{M}\sum_{s \in [M]} \frac{(\sum_{s' \in [M]} p_{s'} B_{s'})^2}{B_s^2})$ goes infinity, it is confirmed. □

### G.2  Proof of Theorem 6

*Proof of Theorem 6.* Since the distribution of $n.$ is invariant against $\theta \in \Theta$, we have

$$\sup_{\theta \in \Theta} \mathbf{E}_\theta\left[\mathcal{E}(\hat{f}_n; \theta)\right]$$

$$= \mathbf{E}\left[\sup_{\theta \in \Theta} \mathbf{E}_\theta\left[\mathcal{E}(\hat{f}_n; \theta)\big|n.\right]\right]$$

$$\geq \mathbf{E}\left[\max_{\theta \in \hat{\Theta}} \mathbf{E}_\theta\left[\mathcal{E}(\hat{f}_n; \theta)\big|n.\right]\right]$$

$$\geq \mathbf{E}\left[\frac{1}{|\hat{\Theta}|} \sum_{\theta \in \hat{\Theta}} \mathbf{E}_\theta\left[\mathcal{E}(\hat{f}_n; \theta) \middle| n_.\right]\right].$$

Given $\epsilon$ possibly dependent on $n_.$, application of the Markov inequality yields

$$\sup_{\theta \in \Theta} \mathbf{E}_\theta\left[\mathcal{E}(\hat{f}_n; \theta)\right]$$

$$\geq \mathbf{E}\left[\frac{\epsilon}{|\hat{\Theta}|} \sum_{\theta \in \hat{\Theta}} \mathbb{P}_\theta\left\{\mathcal{E}(\hat{f}_n; \theta) \geq \epsilon \middle| n_.\right\}\right].$$

If $\inf_f \mathcal{E}(f; \theta) \vee \mathcal{E}(f; \theta') \geq \epsilon$ for any $\theta, \theta' \in \hat{\Theta}$, $\mathcal{E}(f; \theta) < \epsilon$ implies $\mathcal{E}(f; \theta') \geq \epsilon$ for any $\theta' \in \hat{\Theta}$ such that $\theta \neq \theta'$. Hence, there exists a partion $\{\mathcal{F}_\theta\}_{\theta \in \hat{\Theta}}$ of all the measurable functions $f : \mathbb{R}^d \times [M] \to \mathbb{R}$ such that $\{f : \mathcal{E}(f; \theta) < \epsilon\} \subseteq \mathcal{F}_\theta$ for all $\theta \in \hat{\Theta}$. Consequently, we have

$$\sup_{\theta \in \Theta} \mathbf{E}_\theta\left[\mathcal{E}(\hat{f}_n; \theta)\right] \geq \mathbf{E}\left[\epsilon\left(1 - \frac{1}{|\hat{\Theta}|} \sum_{\theta \in \hat{\Theta}} \mathbb{P}_\theta\left\{\hat{f}_n \in \mathcal{F}_\theta \middle| n_.\right\}\right)\right].$$

Application of the Fano's inequality and data processing inequality yields the claim. $\qquad\square$

### G.3 Proof of Theorem 7

To prove Theorem 7, we show the following more tight lower bound.

**Theorem 23.** *Let $\theta$ and $\theta'$ be the parameters of the distributions such that $\frac{1}{2\sigma_X^2}\|\mu_s - \mu_s'\|^2 := d_s < 1$ for all $s \in [M]$. Then, we have*

$$\inf_{f \in \mathcal{L}^2} \mathcal{E}(f; \theta) \vee \mathcal{E}(f; \theta') \geq$$

$$\sum_{s \in [M]} p_s \frac{e^{-d_s}}{4}\left(\sigma_X^2 \left\|\frac{\overline{\|\beta_.\|}\beta_s}{\|\beta_s\|} - \frac{\overline{\|\beta_.'\|}\beta_s'}{\|\beta_s'\|}\right\|^2 \left(1 + \frac{\|\mu_s - \mu_s'\|^2}{4\sigma_X^2}\right)^{1+\frac{d}{2}}\right.$$

$$+ \left(-\left\langle \frac{\overline{\|\beta_.\|}\beta_s}{\|\beta_s\|} + \frac{\overline{\|\beta_.'\|}\beta_s'}{\|\beta_s'\|}, \frac{\mu_s - \mu_s'}{2}\right\rangle\right.$$

$$\left.\left. + \sum_{s' \in [M]} p_{s'}\left(\langle \beta_{s'} - \beta_{s'}', \bar{\mu}_{s'}\rangle + \left\langle \beta_{s'} + \beta_{s'}', \frac{\mu_{s'} - \mu_{s'}'}{2}\right\rangle\right)\right)^2 \left(1 + \frac{\|\mu_s - \mu_s'\|^2}{4\sigma_X^2}\right)^{\frac{d}{2}}\right).$$

Theorem 23 immediately gives Theorem 7.

We utilize the sufficient condition for the constrained optimization problem over a Banach space. Let $Z$ be a Banach space. We say a function $f : Z \to \mathbb{R}$ is *Gateaux differentiable* if the limit $\lim_{\tau \to 0} \frac{f(z+\tau u)-f(z)}{\tau}$ exists for any open set $U \subseteq Z$, any $z \in U$, and any $u \in Z$. We denote the Gateaux derivative of $f$ at $z \in Z$, a linear mapping from $u \in Z$ to $\lim_{\tau \to 0} \frac{f(z+\tau u)-f(z)}{\tau}$, as $D_G f(z)$. We abuse 0 to denote the mapping that always outputs 0.

*Proof of Theorem 23.* Let $q_s$ and $q_s'$ be the density function of $X_s$ with the parameters $(\beta_., \mu_.)$ and $(\beta_.', \mu_.')$, respectively, regarding the base measure $\lambda$. Since $X_s$ follows the Gaussian distribution, we can choose $\lambda$ as the Lebesgue measure. Given $\eta \in [0, 1]$, we have

$$\mathcal{E}(f; \beta_., \mu_.) \vee \mathcal{E}(f; \beta_.', \mu_.')$$

$$\geq \eta \mathcal{E}(f; \beta_., \mu_.) + (1 - \eta)\mathcal{E}(f; \beta_.', \mu_.')$$

$$= \sum_{s \in [M]} p_s \int \left(\eta(f(x, s) - f_{\beta_., \mu_.}(x, s))^2 q_s(x) + (1 - \eta)(f(x, s) - f_{\beta_.', \mu_.'}(x, s))^2 q_s'(x)\right)\lambda(dx).$$

Because $\eta\mathcal{E}(f;\beta_{\cdot},\mu_{\cdot})+(1-\eta)\mathcal{E}(f;\beta'_{\cdot},\mu'_{\cdot})$ is convex for $f$, and $\mathcal{L}^2$ is a Banach space, it is minimized if for any $u \in \mathcal{L}^2$,

$$\frac{d}{d\gamma}\left(\eta\mathcal{E}(f+\gamma u;\beta_{\cdot},\mu_{\cdot})+(1-\eta)\mathcal{E}(f+\gamma u;\beta'_{\cdot},\mu'_{\cdot})\right)\Big|_{\gamma=0}=0.$$

The dominated convergence theorem gives

$$\frac{d}{d\gamma}\left(\eta\mathcal{E}(f+\gamma u;\beta_{\cdot},\mu_{\cdot})+(1-\eta)\mathcal{E}(f+\gamma u;\beta'_{\cdot},\mu'_{\cdot})\right)\Big|_{\gamma=0}$$

$$=\sum_{s\in[M]}p_s\int\Big(\eta(f(x,s)-f_{\beta_{\cdot},\mu_{\cdot}}(x,s))u(x,s)q_s(x)$$

$$+(1-\eta)\big(f(x,s)-f_{\beta'_{\cdot},\mu'_{\cdot}}(x,s)\big)u(x,s)q'_s(x)\Big)\lambda(dx)$$

$$=\sum_{s\in[M]}p_s\int\Big(f(x,s)(\eta q_s(x)+(1-\eta)q'_s(x))$$

$$-\big(\eta f_{\beta_{\cdot},\mu_{\cdot}}(x,s)q_s(x)+(1-\eta)f_{\beta'_{\cdot},\mu'_{\cdot}}(x,s)q'_s(x)\big)\Big)u(x,s)\lambda(dx).$$

Consequently, $\eta\mathcal{E}(f;\beta_{\cdot},\mu_{\cdot})+(1-\eta)\mathcal{E}(f;\beta'_{\cdot},\mu'_{\cdot})$ is minimized at

$$f(x,s)=\frac{\eta f_{\beta_{\cdot},\mu_{\cdot}}(x,s)q_s(x)+(1-\eta)f_{\beta'_{\cdot},\mu'_{\cdot}}(x,s)q'_s(x)}{\eta q_s(x)+(1-\eta)q'_s(x)}.$$

Hence,

$$\eta\mathcal{E}(f+\gamma u;\beta_{\cdot},\mu_{\cdot})+(1-\eta)\mathcal{E}(f+\gamma u;\beta'_{\cdot},\mu'_{\cdot})$$

$$\geq\sum_{s\in[M]}p_s\int\frac{\eta(1-\eta)q_s(x)q'_s(x)}{\eta q_s(x)+(1-\eta)q'_s(x)}\big(f_{\beta_{\cdot},\mu_{\cdot}}(x,s)-f_{\beta'_{\cdot},\mu'_{\cdot}}(x,s)\big)^2\lambda(dx).$$

With $\eta=1/2$, we have

$$\frac{\eta(1-\eta)q_s(x)q'_s(x)}{\eta q_s(x)+(1-\eta)q'_s(x)}=\frac{1}{4}\frac{\sqrt{q_s(x)q'_s(x)}}{\frac{1}{2}\sqrt{\frac{q_s(x)}{q'_s(x)}}+\frac{1}{2}\sqrt{\frac{q'_s(x)}{q_s(x)}}}$$

$$=\frac{1}{4}\frac{\sqrt{q_s(x)q'_s(x)}}{\cosh(\frac{1}{2}\ln\frac{q_s(x)}{q'_s(x)})}.$$

Let $\bar{\mu}_s=\frac{1}{2}(\mu_s+\mu'_s)$. Then, we have

$$\sqrt{q_s(x)q'_s(x)}$$

$$=\frac{1}{\sqrt{(2\pi)^d\sigma_X^{2d}}}\exp\left(-\frac{1}{4\sigma_X^2}\|x-\mu_s\|^2-\frac{1}{4\sigma_X^2}\|x-\mu'_s\|^2\right)$$

$$=\frac{1}{\sqrt{(2\pi)^d\sigma_X^{2d}}}\exp\left(-\frac{1}{2\sigma_X^2}\|x-\bar{\mu}_s\|^2-\frac{1}{2\sigma_X^2}\left\|\frac{\mu_s-\mu'_s}{2}\right\|^2\right.$$

$$\left.-\frac{1}{2\sigma_X^2}\left(\left\langle x-\bar{\mu}_s,\frac{\mu'_s-\mu_s}{2}+\frac{\mu_s-\mu'_s}{2}\right\rangle\right)\right)$$

$$=\frac{1}{\sqrt{(2\pi)^d\sigma_X^{2d}}}\exp\left(-\frac{1}{2\sigma_X^2}\|x-\bar{\mu}_s\|^2-\frac{1}{2\sigma_X^2}\left\|\frac{\mu_s-\mu'_s}{2}\right\|^2\right).$$

Also, we have

$$\frac{q_s(x)}{q'_s(x)}$$

$$= \exp\left(-\frac{1}{2\sigma_X^2}\|x - \mu_s\|^2 + \frac{1}{2\sigma_X^2}\|x - \mu_s'\|^2\right)$$

$$= \exp\left(-\frac{1}{\sigma_X^2}\left\langle x - \bar{\mu}_s, \frac{\mu_s' - \mu_s}{2} - \frac{\mu_s - \mu_s'}{2}\right\rangle\right)$$

$$= \exp\left(\frac{1}{\sigma_X^2}\langle x - \bar{\mu}_s, \mu_s - \mu_s'\rangle\right).$$

Let $\bar{X}_s \sim N(\bar{\mu}_s, \sigma_X^2 I)$. Then, we have

$$\mathcal{E}(f; \beta_\cdot, \mu_\cdot) \vee \mathcal{E}(f; \beta', \mu_\cdot')$$

$$\geq \sum_{s\in[M]} p_s e^{-\frac{1}{2\sigma_X^2}\|\frac{\mu_s - \mu_s'}{2}\|^2} \mathbf{E}\left[\frac{1}{4}\frac{(f_{\beta_\cdot,\mu_\cdot}(\bar{X}_s, s) - f_{\beta',\mu'}(\bar{X}_s, s))^2}{\cosh(\frac{1}{2\sigma_X^2}\langle \bar{X}_s - \bar{\mu}_s, \mu_s - \mu_s'\rangle)}\right]$$

$$\geq \sum_{s\in[M]} p_s e^{-\frac{1}{2\sigma_X^2}\|\frac{\mu_s - \mu_s'}{2}\|^2} \mathbf{E}\left[\frac{1}{4}\frac{(f_{\beta_\cdot,\mu_\cdot}(\bar{X}_s, s) - f_{\beta',\mu'}(\bar{X}_s, s))^2}{\cosh(\frac{1}{2\sigma_X^2}\|\bar{X}_s - \bar{\mu}_s\|\|\mu_s - \mu_s'\|)}\right], \tag{37}$$

where the last line is obtained from the Cauchy–Schwarz inequality.

By definition, we have

$$f_{\beta_\cdot,\mu_\cdot}(x, s) - f_{\beta',\mu'}(x, s)$$

$$= \left\langle \frac{\overline{\|\beta_\cdot\|}\beta_s}{\|\beta_s\|} - \frac{\overline{\|\beta'\|}\beta_s'}{\|\beta_s'\|}, x - \bar{\mu}_s\right\rangle - \left\langle \frac{\overline{\|\beta_\cdot\|}\beta_s}{\|\beta_s\|} + \frac{\overline{\|\beta'\|}\beta_s'}{\|\beta_s'\|}, \frac{\mu_s - \mu_s'}{2}\right\rangle$$

$$+ \sum_{s'\in[M]} p_{s'}\left(\langle\beta_{s'} - \beta_{s'}', \bar{\mu}_{s'}\rangle + \left\langle\beta_{s'} + \beta_{s'}', \frac{\mu_{s'} - \mu_{s'}'}{2}\right\rangle\right).$$

Conditioned on $\|\bar{X}_s - \bar{\mu}_s\| = r$ for $r > 0$, $\bar{X}_s$ follows the uniform distribution over the $(d-1)$-sphere centered at $\bar{\mu}_s$. For a random variable $U$ uniformly distributed over the $(d-1)$-sphere centered at origin with the radius $r$, $\mathbf{E}[U] = 0$ and $\mathbf{E}[UU^\top] = r^2/dI$. Hence, for a vector $v \in \mathbb{R}^d$ and a scalar $c \in \mathbb{R}$, we have

$$\mathbf{E}\left[\left(\langle v, \bar{X}_s - \mu_s\rangle + c\right)^2 \Big| \|\bar{X}_s - \bar{\mu}_s\| = r\right]$$

$$= \frac{r^2}{d}\|v\|^2 + c^2.$$

An elementary analysis yields that $\|\bar{X}_s - \bar{\mu}_s\|^2 \sim \mathrm{Gamma}(\frac{d}{2}, 2\sigma_X^2)$, where $\mathrm{Gamma}(k, \theta)$ denotes the Gamma distribution with the shape parameter $k$ and scale parameter $\theta$. From the upper bound of the hyperbolic cosine as $\cosh(x) \leq e^{x^2/2}$, for a vector $v \in \mathbb{R}^d$, scalars $c \in \mathbb{R}$ and $c' > 0$, and a random variable $\gamma \sim \mathrm{Gamma}(k, \theta)$, we have

$$\mathbf{E}\left[\frac{1}{\cosh(c'\sqrt{\gamma})}\left(\frac{\gamma}{d}\|v\|^2 + c^2\right)\right]$$

$$\geq \mathbf{E}\left[\left(\frac{\gamma}{d}\|v\|^2 + c^2\right)e^{-\frac{c'^2\gamma}{2}}\right]$$

$$= \mathbf{E}\left[\left(\frac{\gamma}{d}\|v\|^2 + c^2\right)\sum_{m=0}^{\infty}(-1)^m\frac{c'^{2m}\gamma^m}{2^m m!}\right]$$

$$= \sum_{m=0}^{\infty}(-1)^m\frac{c'^{2m}}{2^m m!}\left(\frac{\|v\|^2}{d}\theta^{m+1}\frac{\Gamma(k + m + 1)}{\Gamma(k)} + c^2\theta^m\frac{\Gamma(k + m)}{\Gamma(k)}\right)$$

$$= \sum_{m=0}^{\infty}\left(-\frac{c'^2\theta}{2}\right)^m\frac{1}{m!}\left(\frac{k\|v\|^2}{d}\theta\frac{\Gamma(k + 1 + m)}{\Gamma(k + 1)} + c^2\frac{\Gamma(k + m)}{\Gamma(k)}\right)$$

$$= \frac{k\|v\|^2}{d}\theta\left(1 + \frac{c'^2\theta}{2}\right)^{k+1} + c^2\left(1 + \frac{c'^2\theta}{2}\right)^k,$$

where we use the fact that the hypergeometric function $_2F_1(a, b, b; z) = \sum_{m=0}^{\infty} \frac{\Gamma(a+m)}{\Gamma(a)} \frac{z^m}{m!} = (1-z)^{-a}$ for some $b$, provided $|z| < 1$. By setting

$$v = \frac{\overline{\|\beta_{\cdot}\|}\beta_s}{\|\beta_s\|} - \frac{\overline{\|\beta'_{\cdot}\|}\beta'_s}{\|\beta'_s\|}$$

$$c = -\left\langle \frac{\overline{\|\beta_{\cdot}\|}\beta_s}{\|\beta_s\|} + \frac{\overline{\|\beta'_{\cdot}\|}\beta'_s}{\|\beta'_s\|}, \frac{\mu_s - \mu'_s}{2} \right\rangle$$

$$+ \sum_{s'\in[M]} p_{s'}\left( \langle \beta_{s'} - \beta'_{s'}, \bar{\mu}_{s'} \rangle + \left\langle \beta_{s'} + \beta'_{s'}, \frac{\mu_{s'} - \mu'_{s'}}{2} \right\rangle \right)$$

$$c' = \frac{1}{2\sigma_X^2}\|\mu_s - \mu'_s\|$$

$$k = \frac{d}{2}, \text{ and } \theta = 2\sigma_X^2,$$

we have

$$\mathbf{E}\left[ \frac{(f_{\beta_{\cdot},\mu_{\cdot}}(\bar{X}_s, s) - f_{\beta',\mu'}(\bar{X}_s, s))^2}{\cosh(\frac{1}{2\sigma_X^2}\|\bar{X}_s - \bar{\mu}_s\|\|\mu_s - \mu'_s\|)} \right]$$

$$= \sigma_X^2 \left\| \frac{\overline{\|\beta_{\cdot}\|}\beta_s}{\|\beta_s\|} - \frac{\overline{\|\beta'_{\cdot}\|}\beta'_s}{\|\beta'_s\|} \right\|^2 \left( 1 + \frac{\|\mu_s - \mu'_s\|^2}{4\sigma_X^2} \right)^{1+\frac{d}{2}} + \left( -\left\langle \frac{\overline{\|\beta_{\cdot}\|}\beta_s}{\|\beta_s\|} + \frac{\overline{\|\beta'_{\cdot}\|}\beta'_s}{\|\beta'_s\|}, \frac{\mu_s - \mu'_s}{2} \right\rangle \right.$$

$$\left. + \sum_{s'\in[M]} p_{s'}\left( \langle \beta_{s'} - \beta'_{s'}, \bar{\mu}_{s'} \rangle + \left\langle \beta_{s'} + \beta'_{s'}, \frac{\mu_{s'} - \mu'_{s'}}{2} \right\rangle \right) \right)^2 \left( 1 + \frac{\|\mu_s - \mu'_s\|^2}{4\sigma_X^2} \right)^{\frac{d}{2}}.$$

$$(38)$$

Combining Eqs (37) and (38) yields the claim. $\qquad \square$

## G.4  Proof of Theorem 8

*Proof of Theorem 8.* It is easy to check that $d_s = 0$, and for any $v, v' \in \mathcal{V}$,

$$\left\| \frac{\overline{\|\beta_{v,\cdot}\|}\beta_{v,s}}{\|\beta_{v,s}\|} - \frac{\overline{\|\beta_{v',\cdot}\|}\beta_{v',s}}{\|\beta_{v',s}\|} \right\|^2$$

$$= \left( \sum_{s'\in[M]} p_{s'}\|\beta_{v,s}\| \right)^2 \left\| \frac{\beta_{v,s}}{\|\beta_{v,s}\|} - \frac{\beta_{v',s}}{\|\beta_{v',s}\|} \right\|^2$$

$$= \left( \sum_{s'\in[M]} p_{s'}\|\beta_{v,s}\| \right)^2 \left( \sum_{i\in[d-1]} \frac{\epsilon_s^2}{d-1}(v_{s,i} - v'_{s,i})^2 \right)$$

$$= 4\left( \sum_{s'\in[M]} p_{s'}B_s \right)^2 \frac{\epsilon_s^2}{d-1} d_H(v_s, v'_s).$$

Since the density function of the Gaussian distribution is $L^2$ integrable, $\mathcal{E}(f; \theta) = \infty$ if $f$ is not $L^2$ integrable. Hence, $\inf_f \mathcal{E}(f; \theta_v) \vee \mathcal{E}(f; \theta_{v'}) = \inf_{f\in\mathcal{L}^2} \mathcal{E}(f; \theta_v) \vee \mathcal{E}(f; \theta_{v'})$, and we thus can apply Theorem 7. Then, we have

$$\inf_f \mathcal{E}(f; \theta_v) \vee \mathcal{E}(f; \theta_{v'}) \geq \sum_{s\in[M]} p_s \left( \sum_{s'\in[M]} p_{s'}B_s \right)^2 \frac{\sigma_X^2 \epsilon_s^2}{d-1} d_H(v_s, v'_s)$$

Conditioned on $n_{\cdot}$, the KL-divergence between $\pi_{\theta_v|n_{\cdot}}$ and $\pi_{\theta_{v'}|n_{\cdot}}$ is obtained as

$$\sum_{s\in[M]} n_s \left( \frac{1}{2\sigma_X^2}\|\mu_{v,s} - \mu_{v',s}\|^2 + \frac{\sigma_X^2}{2\sigma_\xi^2}\|\beta_{v,s} - \beta_{v',s}\|^2 + \frac{1}{2\sigma_\xi^2}\langle \mu_{v,s}, \beta_{v,s} - \beta_{v',s}\rangle^2 \right).$$

Hence, we have

$$D_{\mathrm{KL}}\big(\pi_{\theta_v|n_\cdot}, \pi_{\theta_{v'},n_\cdot}\big) = \sum_{s\in[M]} \frac{2\sigma_X^2 B_s^2 n_s \epsilon_s^2}{\sigma_\xi^2(d-1)} d_H(v_s, v_s').$$

$\square$

