# OpenReview forum: "Demographic Parity Constrained Minimax Optimal Regression under Linear Model"
_NeurIPS.cc/2023/Conference — NeurIPS 2023 poster_

### Official Review · Reviewer_1oyt · 2023-07-04

**Soundness:** 3 good
**Presentation:** 3 good
**Contribution:** 3 good
**Rating:** 7
**Confidence:** 3

**Summary:**

This paper studies the linear regression problem under a new definition of $(\alpha, \delta)$-fairness consistency as a fairness constraint.
In particular, the authors derived that under the constraint of $(\alpha, \delta)$-fairness, the minimax optimal error is $dM/n$ when $\alpha$ is less than 1/2. Moreover, they provided an estimator that achieves this optimal rate of convergence.

**Strengths:**

This paper established matching upper and lower bounds on the minimax optimal error and proposed a regression algorithm that achieves this optimal error for the linear regression model under a specific fairness constraint. The theory and method extends the current literature to cover partial coefficients and non-sensitive features on the sensitive attribute. The paper is well presented and articulated. The authors also provided detailed proofs which hightlighted the technical difficulties in the minimax analysis, as well as a novel estimator for achieving the minimax optimal bound.

**Weaknesses:**

It would be great if the authors could discuss and provide more understanding on how restrictive the assumption that $\alpha <= 1/2$ is. Also, how should one think about solving for the case that $\alpha > 1/2$?
The paper is lacking empirical study/examples which I feel would be nice given the subject under study.

**Questions:**

It would be great if the authors could discuss and provide more understanding on how restrictive the assumption that $\alpha <= 1/2$ is.
Is it a necessary condition for achieving the minimax rate of dM/n?
Also, how should one think about solving for the case that $\alpha > 1/2$?
How does the fairness score proposed in the paper compare with other fairness scores in the existing literature?
The authors mentioned that misapplication of the findings in the paper to other models might result in discriminatory treatment. Any discussion on the robustness of the proposed algorithms, especially under misspecified models?


**Limitations:**

The authors have pointed out the restriction of the proposed models and cautioned against misusing the algorithms for non-linear models.

---

> ### Author Rebuttal · Authors · 2023-08-07
>
> > It would be great if the authors could discuss and provide more understanding on how restrictive the assumption that $\alpha\le 1/2$ is. Is it a necessary condition for achieving the minimax rate of dM/n? Also, how should one think about solving for the case that $\alpha > 1/2$?
>
> Thank you for raising these important inquiries regarding the condition that $\alpha \le 1/2$. To shed light on your questions, I'd like to begin by assuring you that our lower bound result is indeed applicable to scenarios where $\alpha > 1/2$. Therefore, it should be noted that the minimax rate of dM/n might still be affected even when $\alpha > 1/2$. Furthermore, we should consider the potential for the minimax rate to exceed dM/n if $\alpha > 1/2$.
>
> Diving deeper into the challenges associated with the $\alpha > 1/2$ case, they primarily arise from the necessity to manage the non-linear regressor. Given that the estimation error of the mean of the non-sensitive features stands at the order of $n^{-1/2}$, the outcome distribution similarly deviates at a scale of $n^{-1/2}$ when a linear regression function is implemented. Consequently, achieving an $n^{-\alpha}$ deviation for $\alpha > 1/2$ calls for the use of a non-linear regressor. This, however, complicates the analysis process. We hope this provides a clearer understanding of the issues at hand.
>
> We are currently continuing our research in the case that $\alpha > 1/2$ and have yet to fully understand it. One potential approach to understanding the minimax risk for $\alpha > 1/2$ might be to examine the bias introduced during training. Although our present analyses offer a comprehensive lower bound on the sample bias, there could be additional refinement regarding the bias originating from the training process. We are eager to explore this further in our upcoming work.
>
> > How does the fairness score proposed in the paper compare with other fairness scores in the existing literature?
>
> Thank you for your question regarding the comparison between the fairness score proposed in our study and other fairness scores that are currently available in the literature.
>
> When we consider references [2,7,8], the primary distinction is situated in the selection of the divergence measure. In the context of our work, we have chosen to employ the 2-Wasserstein distance, whereas these studies have favored the use of the Kolmogorov distance. This difference in selection can be associated with the distinct norms embraced in our respective outcome spaces. To clarify, our fairness score is constructed upon the foundation of the $\ell_2$-norm, while their scores are based on the $\ell_\infty$-norm. However, I would like to stress that the underlying principles of these methods are largely congruous.
>
> Moving onto the comparison with the method proposed in reference [6], our score is identified from the pair of groups that exhibits the most pronounced differences. In contrast, their methodology evaluates an average of the divergences of outcome distributions across all groups. This methodological choice inherently gives precedence to the majority group, which might only be suitable in some contexts. We have often noticed, particularly in real-world circumstances, that unfairness tends to be more conspicuous within minority groups. Therefore, our approach aims to address this concern more effectively.
>
> For a more comprehensive comparison, please refer to Appendix A in the supplementary material.
>
> > The authors mentioned that misapplication of the findings in the paper to other models might result in discriminatory treatment. Any discussion on the robustness of the proposed algorithms, especially under misspecified models?
>
> Your viewpoint on the robustness of model misspecification is certainly thought-provoking. We must admit, however, that our current algorithm might not be completely resilient against model misspecification. For instance, if the sensitive attribute influences the variance of non-sensitive features, our algorithm may risk compromising the fairness guarantee. Hence, further investigating the robustness against model misspecification is undeniably an important area for our future work.

---

> > ### Comment · Reviewer_1oyt · 2023-08-16
> >
> > Thank you very much for the detailed response.

---

### Official Review · Reviewer_TKh2 · 2023-07-07

**Soundness:** 4 excellent
**Presentation:** 4 excellent
**Contribution:** 3 good
**Rating:** 7
**Confidence:** 3

**Summary:**

This paper investigates the minimax error in regression under the constraint of demographic parity (DP). In contrast to earlier works, this paper studies both direct and indirect discrimination and their effects on the estimator. In addition to proposing an algorithm to construct a regressor achieving a desirable error, they provide extensive theoretical analysis characterizing upper and lower bounds under the DP constraint.

**Strengths:**

1. The paper is nicely written, with a clear structure and a comprehensive explanation of a large number of notations.

2. It studies a simple but interesting, important problem, which indeed catches attention and makes a good contribution to the community.

3. The introduction of direct and indirect discrimination is novel and realistic, and the examples are helpful for readers to understand those notions.

4. The theoretical analysis is comprehensive. The discussion in Section 4 is interesting to read and may provide insights for future work in this area.

**Weaknesses:**

I did not find particular weaknesses for this paper.

**Questions:**

In your model, you mentioned that indirect discrimination mainly impacts $\mu_S$ while direct discrimination is reflected via $\beta_S^*$. My understanding is that, direct discrimination is for the “model” itself, i.e. $\beta$, while indirect discrimination has influence on the data set. Is my description accurate?

**Limitations:**

The authors addressed them at the end of Section 8.

---

> ### Author Rebuttal · Authors · 2023-08-07
>
> > In your model, you mentioned that indirect discrimination mainly impacts $\mu_S$ while direct discrimination is reflected via $\beta^*_S$. My understanding is that, direct discrimination is for the "model" itself, i.e. $\beta$, while indirect discrimination has influence on the data set. Is my description accurate?
>
> You are largely correct in your understanding. The distinction between direct and indirect discrimination hinges on how the sensitive attribute affects the predicted outcome.
>
> Direct discrimination arises when there's a change in the model parameters due to the sensitive attribute, leading to varying predicted outcomes across different groups. Conversely, indirect discrimination occurs even if the model parameters remain unchanged or independent of the sensitive attribute.
>
> To illustrate using a linear regressor, let's consider $f(x,s) = \mbox{address} \times \beta$, where $\beta$ represents a regression coefficient. At first glance, it might appear that the predicted outcome remains consistent irrespective of the sensitive attribute because $\beta$ is independent of the sensitive attribute. However, if we consider a scenario where the "address" variable is influenced by the sensitive attribute (a phenomenon often seen with race, largely due to historical factors), the predicted outcomes will indeed vary based on the sensitive attribute. We refer to this phenomenon as indirect discrimination.

---

> > ### Comment · Reviewer_TKh2 · 2023-08-16
> >
> > Thanks for the explanation.

---

### Official Review · Reviewer_6KUa · 2023-07-10

**Soundness:** 4 excellent
**Presentation:** 3 good
**Contribution:** 4 excellent
**Rating:** 6
**Confidence:** 3

**Summary:**

The authors study the problem of minimax optimality for the linear regression model under demographic parity constraints from the fairness literature. They provided matching upper and lower bounds for the model that is considered in the paper.

**Strengths:**

Overall:

The paper is well-motivated by presenting both direct and indirect discrimination that occurs in real-world settings and the proposed model accounts for it. The theory is well-laid out and the difficulty is elucidated in the main paper with details left to the appendices. However, there isn't any experimental work.

Pros:

(A) The model differs from the prior literature (Chzhen and Schreuder) and is well motivated in Section 1 and Table 1.
(B) The technical difficulty of both the norm estimators and the direction estimators is presented in equation 7.
(C) The key takeways that the optimal error is independent of mean but depends on the variance is intriguing.


**Weaknesses:**

(1) How does this compare to the results of Chzen and Schreuder is unclear. There is no unified setting which captures both models.
(2) Also, there is no experimental work which would help us know the tightness of the proposed upper bounds (constants).
(3) Some of the writing can be sharpened and there are multiple typos.


**Questions:**

Please check the cons. Slightly lower score because of lack of experimental work.

---

> ### Author Rebuttal · Authors · 2023-08-07
>
> > (1) How does this compare to the results of Chzen and Schreuder is unclear. There is no unified setting which captures both models.
>
> Thank you for drawing attention to the comparison between our work and that of Chzen and Schreuder. Your observation is astute, and we appreciate the opportunity to clarify.
>
> To provide a comparative perspective, we have detailed a side-by-side analysis of our model with that of Chzen and Schreuder's in Lines 77-91. The fundamental difference lies in how each model approaches direct and indirect discrimination.
>
> For direct discrimination, our model is crafted to address disparities originating from the entire regression coefficient. On the other hand, Chzen and Schreuder's model is specialized to address discrimination stemming exclusively from the intercept. When it comes to indirect discrimination, their model, due to its design constraints, doesn't fully capture the indirect discrimination arising from the interdependencies of non-sensitive features with the sensitive attribute. In contrast, our model acknowledges such potential interdependencies, allowing for a more nuanced handling of indirect discrimination through group-dependent means. These distinctions are concisely summarized in Table 1.
>
> You've correctly identified the discrepancy in the settings of our model compared to Chzen and Schreuder's. This distinction primarily arises from the metrics chosen for measuring unfairness. Our method evaluates disparity by focusing on the worst-case pair of groups, whereas Chzen and Schreuder's methodology calculates the average disparity across all groups. Our approach is consistent with prior studies, as seen in references [2,7,8]. An average-based unfairness score might inadvertently favor the majority group, potentially underrepresenting the unfairness faced by minority groups — a situation often observed in real-world contexts. We contend that our method, backed by existing literature, provides a more equitable assessment, specifically reducing undue favoritism towards majority groups.
>
> We are grateful for your thoughtful inquiry and hope this explanation brings added clarity to our approach.
>
>
> > (2) Also, there is no experimental work which would help us know the tightness of the proposed upper bounds (constants).
>
> The core of our paper lies in the theoretical insights related to minimax optimality for demographic parity-constrained regression within a linear model. We are of the strong conviction that these theoretical revelations, irrespective of any accompanying experimental evidence, offer substantial contributions in their own right.
>
> In relation to the multiplicative constant in the upper bounds, we agree that its current value, $C=37044 e^{10}$, may suggest a potential for further refinement. Nonetheless, our study's primary emphasis remains on the exploration of parameters' influence on minimax risk, rendering the precise tightness of the multiplicative constant a secondary concern in this context. Your thoughtful attention to this detail is genuinely appreciated, and we assure you that we will consider delving into this aspect in more detail in our future research endeavors.

---

> > ### Comment · Reviewer_6KUa · 2023-08-22
> >
> > Thanks for clarifying my questions. I will keep my score because of lack of experimental work and the high constants in the bound. However, it is a good theoretical contribution.

---

### Official Review · Reviewer_okbq · 2023-07-26

**Soundness:** 4 excellent
**Presentation:** 4 excellent
**Contribution:** 4 excellent
**Rating:** 7
**Confidence:** 4

**Summary:**

This paper studies the statistical minimax error of building a linear regression model under the constraint of demographic parity.
It generalizes the result in Chzhen and Schreuder [5] by allowing the slope coefficients to change with sensitive attribute rather than just
the intercepts. The authors provide both upper and lower bounds of the minimax error, and show that they match in the dependency on $n$.

**Strengths:**

Although this paper uses a framework similar to  Chzhen and Schreuder [5], it covers a very important setting where there exists both direct and indirect discriminations. The analysis begins with Lemma 1, which can be easily derived from Chzhen and Schreuder [5]. However, to obtain the upper bound of the minimax error, the authors need to characterize the errors of approximating each component in (5), which is one of the main challenges in this paper. They develop new techniques in Theorems 4 and 5 to handle this challenge. I think it has made significant contribution on top of   Chzhen and Schreuder [5].

**Weaknesses:**

1. This paper assumes the covariance matrix of X is an identity matrix for each $s$. It will be better to allow a more general matrix $\Sigma_s$.
2. It only covers the case where $\alpha\leq 1/2$.

**Questions:**

1. The statement made in line 136-137 should be explained more clearly. In particular, why does enforcing strict demographic parity result in a constant function?

2. The authors consider $(\alpha, \delta)$-consistently fair regressor while Chzhen and Schreuder [5] consider $(\alpha, t)$-valid estimators (see DEFINITION 5.2 in [5]). Are they equivalent? If not, why do the authors consider a different type of fair regressor in the definition of the minimax error.

3. Equation (4) needs to be explain, especially the second inequality. Also, I think $\beta$ here should be $\beta^*$.

**Limitations:**

See weaknesses above.

---

> ### Author Rebuttal · Authors · 2023-08-07
>
> > The statement made in line 136-137 should be explained more clearly. In particular, why does enforcing strict demographic parity result in a constant function?
>
> Thank you for your query regarding the explanation provided in lines 136-137. The occurrence of this particular phenomenon can be better understood by considering the learning algorithm's limited ability to access the underlying distribution. Here, we can conceptualize the learning algorithm as a mapping from a sample $D_n$ to a regressor $\hat{f}_n$. It is important to understand that, given a particular sample, the algorithm is designed to yield a particular regressor.
>
> For a specific sample, $D_n$, we see that the density it perceives is non-zero across all underlying distributions. This is due to the inherent nature of a Gaussian distribution, which, irrespective of its parameters, is supported on the entire real vector space. Despite the probability being small, there is still the potential for a specific sample, $D_n$, to be observed across all underlying distributions.
>
> Viewing the situation from this perspective, it is clear that a corresponding regressor $\hat{f}_n$, learned from $D_n$, must adhere to the strict demographic parity constraint across all distributions. This obligation is based on the premise that any given distribution has a non-zero likelihood of producing the sample $D_n$. This understanding can be generalized to all samples that are processed by the algorithm. Thus, it follows that the algorithm should generate a regressor that upholds the strict demographic parity constraint for every underlying distribution. The sole function that consistently satisfies this stringent demographic parity constraint is a constant function. We hope this elaboration addresses your query satisfactorily.
>
> > The authors consider $(\alpha,\delta)$-consistently fair regressor while Chzhen and Schreuder [5] consider $(\alpha,t)$-valid estimators (see DEFINITION 5.2 in [5]). Are they equivalent? If not, why do the authors consider a different type of fair regressor in the definition of the minimax error.
>
> Thank you for drawing a comparison between our proposed $(\alpha,\delta)$-consistently fair regressor and the $(\alpha,t)$-valid estimators as defined by Chzhen and Schreuder. While these two concepts are interconnected, they are not equivalent and possess nuanced differences. To clarify the difference between these definitions, we use $(\gamma, t)$-validness to represent Chzhen and Schreuder's definition.
>
> Specifically, when we choose an appropriate unfairness score, an $(\alpha,\delta)$-consistently fair regressor transforms into a $(Cn^{-\alpha},\delta)$-valid entity for some constant $C > 0$. Similarly, a $(Cn^{-\alpha},t)$-valid estimator, given a particular $C > 0$, can be redefined as an $(\alpha,t)$-consistently fair entity.
>
> The distinguishing factor here is the treatment of $\gamma$: while the $(\gamma,t)$-validity allows $\gamma$ to remain constant, our proposed concept of consistent fairness necessitates a diminishing $\gamma$ as $n$ increases. Although our definition of fairness consistency may be more constrained than that proposed by Chzhen and Schreuder, the feature of converging towards a strictly fair regressor is a favorable attribute in practical applications.
>
> Additionally, it is pertinent to mention the disparity in the unfairness scores we use compared to Chzhen and Schreuder. Our approach measures the disparity between the worst-case pair of groups, whereas their method averages the disparity across all groups. Our method of measurement is not unique but is endorsed by other researchers as well, as demonstrated in [2,7,8] where similar unfairness scores have been used. In comparison, the average-type unfairness score [6] could unintentionally favor the majority group, which may not always be an optimal approach as it often overlooks the greater unfairness experienced by minority groups. This is an important consideration as such situations frequently occur in real-life contexts.
>
> > Equation (4) needs to be explain, especially the second inequality.
>
> Thank you for your inquiry regarding Equation (4), particularly the second inequality. We'd be pleased to provide further clarification.
>
> The second inequality found in Equation (4) illustrates the variability of the norms $\|\beta^*_s\|$ under the restriction set forth by the first inequality. More specifically, a higher value for $B$ allows a more diverse values for the norm $\|\beta^*_s\|$.
>
> We can interpret the left-hand side of the second inequality in Equation (4) as a product of two factors: the weighted average norms, $\sum_sp_s|\beta^*_s|$, and the averaged inverse norms, expressed as $\frac{1}{M}\sum_s\frac{1}{\|\beta^*_s\|}$. As the norms increase, the first factor (weighted average norms) has the propensity to grow, while the second factor (averaged inverse norms) tends to rise when the norms decrease. Maximizing the product of these two elements involves a delicate balancing act: the norms of some groups need to be large, while the norms of other groups need to be smaller. As such, the left-hand side of the second inequality can increase when the norms $|\beta^*_s|$ display diversity. I hope this explanation clarifies your doubts.

---

### Author Rebuttal · Authors · 2023-08-07

We would like to express our gratitude to all the reviewers for dedicating their time to providing such careful reviews. We also greatly appreciate the high evaluations that our papers received from the reviewers. Please refer to the individual responses for our answers to the questions the reviewers raised.

---

### Decision · Program_Chairs · 2023-09-21

**Decision:**

Accept (poster)

**Comment:**

The paper extends the Chzhen and Schreuder's framework to address both direct and indirect discrimination in linear regression models under demographic parity. It innovatively establishes matching upper and lower bounds on the minimax optimal error through theorems and introduces an algorithm that achieves this optimal error. While the work is theoretically strong, it lacks empirical validation as no experimental results are included. I suggest the authors consider adding some experiments if possible.